# Quantitative Mass Spectrometry Characterizes Client Spectra of Components for Targeting of Membrane Proteins to and Their Insertion into the Membrane of the Human ER

**DOI:** 10.3390/ijms241814166

**Published:** 2023-09-15

**Authors:** Martin Jung, Richard Zimmermann

**Affiliations:** Medical Biochemistry and Molecular Biology, Saarland University, 66421 Homburg, Germany; martin.jung@uks.eu

**Keywords:** gene expression, protein biogenesis, membrane proteins, endoplasmic reticulum, membrane targeting, membrane insertion, signal recognition particle, Sec61 complex

## Abstract

To elucidate the redundancy in the components for the targeting of membrane proteins to the endoplasmic reticulum (ER) and/or their insertion into the ER membrane under physiological conditions, we previously analyzed different human cells by label-free quantitative mass spectrometry. The HeLa and HEK293 cells had been depleted of a certain component by siRNA or CRISPR/Cas9 treatment or were deficient patient fibroblasts and compared to the respective control cells by differential protein abundance analysis. In addition to clients of the SRP and Sec61 complex, we identified membrane protein clients of components of the TRC/GET, SND, and PEX3 pathways for ER targeting, and Sec62, Sec63, TRAM1, and TRAP as putative auxiliary components of the Sec61 complex. Here, a comprehensive evaluation of these previously described differential protein abundance analyses, as well as similar analyses on the Sec61-co-operating EMC and the characteristics of the topogenic sequences of the various membrane protein clients, i.e., the client spectra of the components, are reported. As expected, the analysis characterized membrane protein precursors with cleavable amino-terminal signal peptides or amino-terminal transmembrane helices as predominant clients of SRP, as well as the Sec61 complex, while precursors with more central or even carboxy-terminal ones were found to dominate the client spectra of the SND and TRC/GET pathways for membrane targeting. For membrane protein insertion, the auxiliary Sec61 channel components indeed share the client spectra of the Sec61 complex to a large extent. However, we also detected some unexpected differences, particularly related to EMC, TRAP, and TRAM1. The possible mechanistic implications for membrane protein biogenesis at the human ER are discussed and can be expected to eventually advance our understanding of the mechanisms that are involved in the so-called Sec61-channelopathies, resulting from deficient ER protein import.

## 1. Introduction

Nucleated human cells are separated from the environment by the so-called plasma membrane and contain different subcellular compartments, called cell organelles (Figure 1). These organelles are surrounded and, thereby, separated from the aqueous, albeit gel-like, cytosol by at least one biological membrane and have to be distributed to daughter cells from the mother cell during cell division (with the exception of lipid droplets and peroxisomes). In the cytosol, the vast majority of the approximately 24,000 different polypeptides of human cells are synthesized by 80S ribosomes. Therefore, the distinct proteins of the various organelles and the plasma membrane have to, first, be targeted to the specific organelles and, subsequently, inserted into or translocated across the membrane(s) of the relevant organelles. The protein import into the organelle termed the endoplasmic reticulum (ER) is the first step in the biogenesis of about one-third of the different soluble and membrane proteins (MPs) of human cells and, therefore, represents a central cell biological research topic of the past fifty years as well as several years to come. In a second step, the non-ER proteins reach their functional location in either the extracellular space; one of the endocytotic or exocytotic organelles (ERGIC, Golgi apparatus, endosome, lysosome, or trafficking vesicles); the plasma, peroxisomal, and mitochondrial membrane or in lipid droplets by vesicular transport; direct budding of new organelles (peroxisomal precursors or lipid droplets); or the ER–SURF pathway (mitochondria) [1,2,3,4,5,6,7,8,9,10]. The first insights into ER protein import were gained about seventy years ago. From electron microscopic images, Palade and Potter concluded that the ER represents a ‘continuous, tridimensional reticulum’ and that ‘the surface appears to be dotted with small, dense granules that cover them in part or in entirety’, i.e., cytosolic 80S ribosomes [11,12].

The latter observation paved the way for the ‘signal hypothesis’ by G. Blobel and colleagues [1,2,13,14], which suggested that topogenic sequences in nascent precursor polypeptides guide the translating ribosomes to the ER membrane and that the subsequent membrane insertion or translocation occurs coupled to translation, i.e., cotranslationally (Figure 1). For membrane proteins, the beauty of this concept of cotranslational ER protein import is that their eventual hydrophobic transmembrane domain or domains do not face the problem of aggregation in the cytosol. Subsequent work in human cell-free systems and in the yeast *Saccharomyces cerevisiae* uncovered that the topogenic sequence, termed the amino-terminal signal peptide (SP) or SP-equivalent transmembrane helix (TMH), of a nascent precursor polypeptide is recognized and bound by the cytosolic signal recognition particle (SRP), which facilitates the association of the complex between the ribosome, nascent chain, and SRP with the heterodimeric SRP receptor in the ER membrane, termed the SR (Figure 2 and Figure 3) [2,15,16,17,18,19,20,21,22,23,24,25,26,27,28]. Thus, the combined action of SRP plus SR represents an ER targeting pathway for nascent precursors of soluble and membrane proteins, as well as the corresponding mRNAs. Recently, proximity-based ribosome-profiling experiments confirmed the preference of SRP and SR for SPs and relatively amino-terminal TMHs of the nascent precursor polypeptide chains [29,30]. Typically, the interaction of SRP with SR leads to the cotranslational transfer of the ribosome-nascent chain complex (RNC) to the central component for both protein translocation and membrane insertion in the ER membrane, the translocon, or the heterotrimeric Sec61 complex (Figure 1) [31,32,33,34,35,36,37,38,39,40,41]. SPs and TMHs of nascent precursors may spontaneously interact with and trigger the opening of the Sec61 channel; i.e., both the central aqueous channel, as well as the lateral gate (Figure 1), or the productive Sec61 interaction may be facilitated by one of the auxiliary components of the ER membrane, i.e., the heterotetrameric translocon-associated protein (TRAP) [42,43,44,45,46,47,48], the heterodimeric Sec62/Sec63 complex with or without the help of the ER lumenal chaperone BiP [49,50,51,52,53,54,55,56,57,58,59,60,61,62], and the translocating chain-associated membrane protein 1 (TRAM1) (Figure 1 and Figure 3) [63,64,65,66,67]. Notably, however, there is a human paralog of the α-subunit of the Sec61 complex, termed Sec61α2, i.e., a putative alternative Sec61 complex that is more or less uncharacterized but recently addressed with respect to its substrates or client spectrum in yeast [68], and there exist alternative components for the targeting (SND, TRC/GET, and PEX3) [69,70,71,72,73,74,75,76,77,78,79,80,81,82,83,84,85,86,87,88,89,90,91,92,93,94,95,96,97,98,99,100,101,102,103,104,105,106,107,108,109,110,111,112,113,114,115,116,117,118,119,120], as well as membrane insertion of precursors (EMC and the GEL-BOS-PAT complex) [121,122,123,124,125,126,127,128,129,130,131,132].

**Figure 2 ijms-24-14166-f002:**
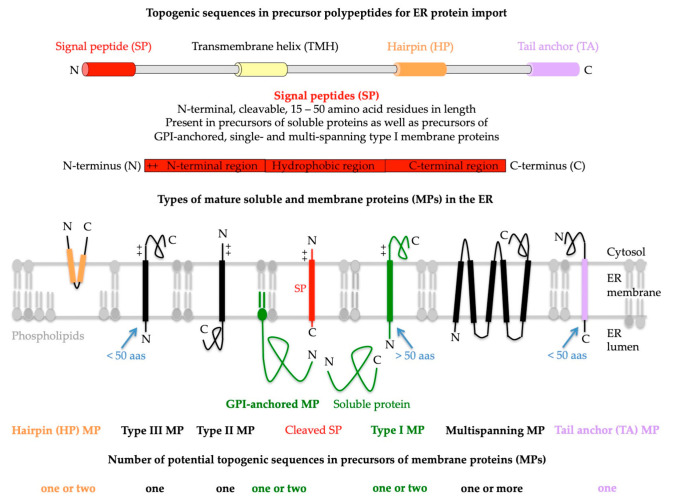
Topogenic sequences in precursors of soluble and membrane proteins for import into the human endoplasmic reticulum (ER). The cartoons depict signal peptides (SPs, shown in red) and seven classes of ER membrane proteins (MPs), plus their membrane protein types (in bold face) and potential topogenic sequences. Cleavable SPs have a tripartite structure (including a hydrophobic or H-region with 7–9 amino acid residues, or aas for short) and facilitate ER import of soluble proteins, GPI-anchored MPs, and single-spanning type I MPs (all shown in green). In addition, they mediate ER import of some multispanning MPs, but not of hairpin, single-spanning type II or III, TA, and other multispanning MPs (which can also be of type II or III; by definition, the shown multispanning MP is of type III). All the latter MPs depend on more or less amino-terminal transmembrane helices (TMHs, with a length of 15–25 amino acid residues) that serve as SP equivalents and facilitate membrane targeting, as well as membrane insertion. Notably, precursors of soluble proteins with SP and of type II or III and TA MPs contain only one topogenic sequence, whereas, in all other precursors, other TMDs than the TMHs may serve as alternative or additional topogenic sequences. Positively charged amino acid residues (+) play an important role in the orientation of SPs and MPs in the ER membrane, where the orientation follows the positive inside rule. Cleavable SPs are removed from the precursor polypeptides in transit by one of the two signal peptidase complexes (SPCs), which have their catalytic sites in the ER lumen. Following ER import and simultaneous cleavage of the SP, as well as the C-terminal GPI-attachment sequence, GPI-MPs become membrane-anchored via carboxy-terminal GPI attachment. Notably, HP proteins are also termed monotopic MPs, and type II and type III MPs are also referred to as MPs with N_in_ or N_cyt_ and N_out_ and N_exo_, respectively, signal anchors [1,69,70]. The figure and its legend were adapted from Lang et al. and Sicking et al. [5,6]. C, carboxy-terminus; GPI, glycosylphosphatidylinositol; N, amino-terminus.

### 1.1. Proteins of the ER Membrane

The SPs, mentioned above, have a tripartite structure (including an amino-terminal or N-region, a hydrophobic or H-region, and a carboxy-terminal or C-region) and facilitate the ER targeting of soluble proteins, GPI-anchored MPs, and single- as well as multispanning type I MPs (Figure 2) [15,16,17,69,70]. In contrast, all the other MPs of the ER membrane (hairpin or HP proteins, and single- and multispanning type II or type III MPs, as well as TA proteins) depend on more or less amino-terminal or even carboxy-terminal TMHs (with a typical length of 15–25 amino acid residues) that serve as SP equivalents and facilitate membrane targeting as well as membrane insertion [15,16,17]. HP- and GPI-anchored proteins are special [71,72,73,74,75]: Following ER import and the simultaneous cleavage of the SP, as well as the carboxy-terminal GPI-attachment sequence, GPI-anchored MPs become membrane-anchored via carboxy-terminal GPI attachment. HP proteins do not contain a real transmembrane domain (TMD); instead, they comprise one or more hydrophobic stretches of amino acid residues, which are typically defined by databases as TMDs and, therefore, discussed here as such. By definition, precursors of soluble proteins with SP and of single-spanning type II or III and TA MPs contain only one topogenic sequence, whereas, in all other precursors, other TMDs (i.e., other than the most amino-terminal TMHs) may serve as alternative or additional topogenic sequences. Typically, SPs are removed from the precursor polypeptides in transit by signal peptidases with ER lumenal catalytic sites [76,77,78]. Furthermore, many of the polypeptides of the secretory pathway become N-glycosylated by one of the two oligosaccharyltransferases (OST), which also have their catalytic sites in the ER lumen but act either on precursors in transit (i.e., cotranslationally, OSTA, Figure 1) or posttranslationally (OSTB) [79,80,81].

### 1.2. Targeting of Precursor Polypeptides to the ER Membrane

The characterization of precursors capable of SRP-independent ER targeting, such as small presecretory proteins and TA proteins, first suggested the existence of SRP-independent ER targeting pathways to the ER [82,83,84,85,86,87]. In contrast to the SRP/SR pathway, these alternative targeting pathways can direct precursor polypeptides to the Sec61 complex co- as well as posttranslationally, and are named the TRC or GET, PEX19/PEX3, and Snd2/Snd3 pathway (Figure 3) [58,71,73,74,75,87,88,89,90,91,92,93,94,95,96,97,98,99,100,101,102,103,104,105,106]. Notably, there is also SRP-independent targeting of all sorts of mRNAs to the ER surface (Figure 3) [29,107,108,109,110]. Typically, this mRNA targeting to the ER depends on receptors for mRNAs (such as KTN1), or RNCs with nascent polypeptide chains that are not yet long enough to allow the interaction of the topogenic sequence with SRP (such as RRBP1, LRRC59, and AEG-1) (Figure 3) [29,111,112,113]. In the case of the ER targeting of mRNAs that code for cytosolic proteins, the nascent polypeptide-associated complex (NAC) can bind to the amino-terminus of the nascent polypeptides and trigger their release from the Sec61 complex [23,114,115,116].

### 1.3. Insertion of Precursor Polypeptides into the ER Membrane

As stated above, the insertion of precursors with SP or TMH into the Sec61 channel and the concomitant gating of the Sec61 channel to the open conformation occur spontaneously or involve client-specific auxiliary components of the Sec61 channel (TRAP, Sec62/Sec63 complex) (Figure 1 and Figure 3). Typically, the orientation of SP- and TMH in the Sec61 channel follows the positive inside rule; i.e., positively charged amino acid residues in the N-region support loop insertion (N_in_-C_out_) and positively charged residues downstream of the SP or TMH interfere with loop insertion and, therefore, favor head-on insertion (N_out_-C_in_) into the Sec61 channel (Figure 2) [117,118,119,120]. Next, TMDs can enter the phospholipid bilayer via the lateral gate of the fully open Sec61 channel by lateral movement and large hydrophilic domains (i.e., with more than 50 amino acid residues), or entire soluble proteins can be translocated into the ER lumen by vectorial movement through the channel (Figure 1 and Figure 2). Alternatively, membrane insertion of some precursors of MPs can be facilitated co- or posttranslationally by evolutionarily conserved MP insertases that comprise a hydrophilic vestibule near the cytosolic surface of the ER membrane, but cannot form an aqueous channel through the ER membrane, and, therefore, cannot translocate hydrophilic domains with more than 50 amino acid residues across the ER membrane. These Oxa1-related insertases are the Wrb/Caml complex (in co-operation with cytosolic GET3 of the abovementioned TRC pathway), or the multimeric ER membrane protein complex (EMC, which may occasionally co-operate with the Sec61 complex), or the GEL-BOS-PAT complex, which depends on the interaction with but not the channel activity of the Sec61 complex (Figure 3) [121,122,123,124,125,126,127,128,129,130,131,132]. Notably, the first has its exclusive role in the membrane insertion of TA membrane proteins [92,95,98]. In the case of the EMC, a proteomic approach identified TA proteins, as well as multispanning MPs, as the predominant clients, the latter of which were also characterized as the main clients of the GEL-BOS-PAT-Sec61 supercomplex but proposed to have a preference for multispanning MPs with four or more TMDs, including those with marginal hydrophobicity [131,132]. There are two excellent reviews, plus several very recent original articles, on the structural and mechanistic details of the various MP insertases in the ER membrane [69,70,81,121,122,123,124,125,126,127,128,129,130,131,132]. The focus of this review, however, is on the client profiles of the various membrane targeting and insertion components and their overlaps in human cells under in-vivo-like conditions, i.e., in intact human cells, as outlined next.

### 1.4. A Single Proteomic Approach to Address the Client Spectra of Various Components for Targeting of Precursor Polypeptides to and Insertion into the Human ER Membrane

Following the pioneering work by G. Blobel and B. Dobberstein [13,14], protein import into the human ER was usually studied with the focus on single precursor polypeptides that were analyzed one-by-one in cell-free assays or intact cells. These classical studies led to the conclusions of whether and how the targeting and membrane insertion or translocation of a certain precursor is facilitated by a certain component. Recently, more global approaches were developed, such as the already mentioned proximity-specific ribosome-profiling [29,111,126,133,134] and quantitative proteomics approaches [5,106,112,126,135,136,137,138,139]. Typically, the proteomic approach employed siRNA-mediated knock-down or CRISPR/Cas9-mediated knock-out of components one-by-one in human cells, label-free quantitative proteomic analysis, and differential protein abundance analysis to characterize client specificities of components. In contrast to the classical analyses, the quantitative proteomics approach addresses the question of which precursors use a certain pathway or component in intact cells, i.e., under in-vivo-like conditions. Previous comparative analyses were carried out for SP-containing precursor polypeptides and their membrane targeting, as well as translocation, and, in the case of membrane protein precursors, for targeting components [5,106]. Here, we extended the analysis to components for membrane insertion and performed a comparative analysis to reach a first and almost complete comprehensive summary.

## 2. Results

### 2.1. A Novel Proteomic Approach for the Analysis of Protein Import into the Human ER

As previously outlined [5], ‘the approach relies on the fact that precursors polypeptides are degraded by the cytosolic proteasome upon interference with their ER import. Therefore, their cellular levels are negatively affected compared to control cells. This change is detected by quantitative MS in combination with differential protein abundance analysis [135]. The depletion of Sec61α served as a proof of principle [135], but, meanwhile, the approach was extended to almost all other ER protein import components [106,112,126,135,136,137,138,139]. Typically, 5000 to 6500 different proteins were quantified and statistically analyzed. For the control cells, gene ontology (GO) terms assigned the expected 27.5% of proteins to organelles of the endocytic and exocytic pathways, plus cell surface [5]. In the case of the depletion or deficiency of an ER protein import component, GO terms typically assigned 35 to 60% of the negatively affected proteins to organelles of the pathways of endocytosis and exocytosis, representing a more or less pronounced enrichment. Typically, a similar enrichment of precursor proteins with SP, N-glycosylation, or membrane location was observed. Taken together, these results support the idea that the precursors of these negatively proteins are potential clients of the components of interest. Notably, however, our experimental approach underestimates the number of different precursor polypeptides relying on this component by far’.

The clients were characterized with various online tools to identify the client spectra of the different components and to deduce the rules of their engagement [135]. Interestingly, precursors with below-average hydrophobic SPs were found to be more strongly affected by Sec61 depletion. Thus, precursors with a higher SP hydrophobicity appear to be more efficient in triggering Sec61 channel opening than those with lower hydrophobicity, which may be related to the hydrophobic patch formed by four residues of Sec61α TMDs 2 and 7 that line the lateral gate of the channel and were found to be important for its opening [39,40]. Notably, SP hydrophobicity was also observed to be key to the roles of two auxiliary components of the Sec61 channel, TRAP and Sec62/Sec63, which may explain why the two auxiliary complexes showed some client overlaps [135,136]. For SPs having a low overall hydrophobicity in combination with high GP content and, therefore, low alpha-helical propensity, full Sec61 channel opening in cotranslational transport was found to be supported by TRAP [135]. On the other hand, the low H-region hydrophobicity of the SPs, particularly in combination with detrimental features within the mature part of the clients, was found to require support from Sec62/Sec63 with or without additional BiP involvement for full Sec61 channel opening in co- as well as posttranslational transport [136]. When the TMHs of the precursors of MPs were analyzed, however, for hydrophobicity, GP content, and ΔG for membrane insertion, no significant distinguishing features were determined [135,136]. Therefore, additional and alternative features of MPs had to be considered in the cases of precursors with TMH. The original MS data were deposited to the ProteomeXchange Consortium (http://www.proteomexchange.org, accessed on 1 April 2023) with the dataset identifiers given in Appendix A.

### 2.2. Precursor Polypeptide Targeting to the Human Endoplasmic Reticulum

The SRP-independent or SND pathway was discovered in a high-throughput screening approach in yeast [74]. In contrast to the SRP/SR system, the SND pathway in yeast showed a preference for clients with central or even carboxy-terminal TMHs (including TAs as well as GPI-attachment sequences of GPI-anchored proteins), rather than amino-terminal TMHs. Furthermore, the SND pathway was able to provide an alternative targeting route for clients with TMHs at their amino-terminus. This work identified the ER membrane protein TMEM208 as human Snd2 ortholog (named Snd2) [74] and TMEM109 as putative human Snd3 homolog [74,106]. In experiments combining siRNA-mediated gene silencing with protein transport into the ER of semi-permeabilized cells in cell-free assays, Snd2 and Snd3 showed a similar function as their yeast counterparts, i.e., facilitated the targeting of TA membrane proteins, as well as small presecretory proteins, to the ER [57,102,103,106]. Recently, precursors of multispanning MPs and of various GPI-anchored proteins were added to the list of human SND clients [104,105]. In contrast, HP and TA proteins typically involve other components and posttranslational pathways for their ER import (Figure 3). The TRC or GET pathway targets to the ER and inserts into the ER membrane TA proteins, and the PEX3-dependent pathway targets and, possibly, inserts HP proteins and certain peroxisomal membrane proteins (Figure 2) [71,73,92,95,98]. In the case of the TRC pathway, membrane targeting involves the cytosolic Bag6 complex, as well as additional cytosolic and ER membrane resident components (Wrb/Caml); in the case of the PEX3-dependent pathway, membrane targeting involves cytosolic PEX19 and the ER membrane protein PEX3, or, speculatively, a heterodimeric complex of PEX3 and PEX16 (Figure 3). Apparently, these three SRP-independent targeting pathways are not fully separated; i.e., some small presecretory proteins with a content of less than 100 amino acid residues can be posttranslationally targeted to the Sec61 channel by the SRP, SND, as well as the TRC pathway [57,58,103]. Likewise, some TA membrane proteins can be posttranslationally targeted to the ER membrane via the same three pathways [101]. Thus, there is redundancy in the targeting process; i.e., the targeting pathways have overlapping client specificities. As stated above, the TRC or GET pathway and the PEX19/PEX3 pathway do not only facilitate the targeting of precursors to the ER membrane but also mediate the insertion of one class of MP precursors into the ER membrane, the TA proteins in the case of TRC and the HP proteins in the case of PEX19/PEX3. The underlying mechanisms are special, i.e., distinct from the mechanism of membrane insertion of MPs via the Sec61 complex, because they can transfer across the membrane only short (with a content of less than 50 amino acid residues, Figure 2) or no hydrophilic domains, and are, in part (Wrb), evolutionarily related to the mitochondrial Oxa1 protein and subunits of two additional membrane protein insertases (EMC and GEL/BOS complex) which are located in the ER membrane.

#### 2.2.1. The SRP/SR Targeting Pathway

As described in the introduction, the SRP/SR pathway represents the archetype cotranslatioal targeting mechanism, delivering nascent precursor polypeptide chains early in their synthesis to the insertion and translocation machinery in the target membrane [18,19,20,21,22,23,24,25,26,27,28]. As previously reported [106], the SRα depletion caused the degradation of the second subunit of the SRP receptor, SRβ. Therefore, the cells were actually depleted of the entire SR complex. The SR client spectrum comprised 44.4% precursors with SP (including 25.9% precursors of type I MPs but no multispanning MPs) and 55.6% precursors of MPs with TMH, including the peroxisomal MP PEX3 (Figure 4A, Appendix A). Thus, 81.5% of the SR clients were MPs. The precursors with TMH included 31.5% multispanning MPs, 18.5% single-spanning type II and type III MPs, and 5.6% HP proteins (Figure 4A, Appendix A). As expected, there were no TA proteins found among the SR clients. When the TMHs were analyzed for their positioning within the precursor proteins, 77% turned out to have comparatively N-terminal TMHs (Figure 5A, Appendix A). Thus, the SR client spectrum demonstrated a clear overall preference for amino-terminal topogenic sequences (87%, i.e., 24 with SP, plus 23 with relatively amino-terminal TMH, out of the total of 54 clients, according to Appendix A), which is consistent with the global ribosome-profiling experiments [30]. In addition, an overall preference of SR (31.5%) for multispanning MPs became apparent (Figure 4A, Appendix A). Furthermore, the presence of the peroxisomal PEX3 among the SR clients confirmed the feasibility of the experimental approach [140].

#### 2.2.2. The SRP-Independent or SND Targeting Pathway

The SND system can act posttranslationally and, therefore, accept small presecretory proteins and small or TA-MPs, as well as GPI-anchored MPs, as clients [57,58,74,75,101,103,104,105,106]. Therefore, some of these clients also involve Sec62, an auxiliary component of the posttranlationally acting translocon [57,58,75]. The SND client spectrum were as follows: 29.1% precursors with SP (including 14.5% precursors of type I MPs, plus 3.6% precursors of multispanning MPs) and 70.9% precursors of MPs with TMH (Figure 4B, Appendix A). Thus, 89% of the SR clients were precursors of MPs. The precursors with TMH included 47.3% multispanning MPs, 9.1% single-spanning type II and type III MPs, 12.7% TA proteins, and 1.8% HP proteins (Figure 4B, Appendix A). Thus, the SND client spectrum showed a clear preference of 50.9% for multispanning MPs, as compared to 31.5% for SR. When the TMHs were analyzed for their positioning within the precursor proteins, 42% turned out to have central or even C-terminal TMHs (Figure 5B, Appendix A), as compared to 24% for SR. Overall, the SND client spectrum demonstrated a lower preference for amino-terminal topogenic sequences (71%) as compared to SR (87%), which is consistent with the observations for this pathway in yeast cells [74]. Furthermore, a pronounced overall preference of SND (40.9%) for multispanning MPs became apparent (Figure 4B, Appendix A).

#### 2.2.3. The TRC/GET Pathway

Although it also includes ribosome-associated components, the TRC/GET system was expected to act posttranslationally by definition since it can only come into action after most of the precursor has already been synthesized. i.e., because of the carboxy-terminal topogenic sequence [87,92,98]. Therefore, it can also accept small presecretory proteins as clients [57,58,101]. Furthermore, it can also target HP proteins to the ER in yeast [141].

As previously reported, the Wrb depletion caused the degradation of the second subunit of the Wrb/Caml complex, as well the cytosolic components of the pathway, TRC35 and TRC40 [106]. Thus, the cells were depleted of the entire pathway. Here, the TRC/GET client spectrum are as follows: 37.1% precursors with SP (including 15.7% precursors of type I MPs, plus 2.9% precursors of multispanning MPs) and 62.9% precursors of MPs with TMH, including the peroxisomal TA protein FAR1 (Figure 4C, Appendix A). Thus, 81.4% of the TRC/GET clients were precursors of MPs. The precursors with TMH included 41.4% multispanning MPs, 11.4% TA proteins, 7.1% type II and type III membrane, and 2.9% HP proteins (Figure 4C, Appendix A). Thus, the TRC/GET client spectrum showed a slight preference of 44.3% for multispanning MPs, as compared to 31.5% for SR and 50.9% for SND. When the TMHs were analyzed for their positioning within the precursor proteins, 47% turned out to have central or even C-terminal TMHs (Figure 5C, Appendix A), as compared to 24% for SR and 42% for SND. Overall, the TRC/GET client spectrum demonstrated a lower preference for N-terminal topogenic sequences (70%) as compared to SR (87%), and a similar one as compared to SND (71%). Furthermore, a surprising overall preference of Wrb for multispanning MPs emerged from these results (44.3%) [106], which is comparable to the overall preference of SND (40.9%) (Figure 4B,C, Appendix A). These results suggested a more general targeting role of the TRC pathway than previously anticipated. However, first hints in this direction had previously come from the observation that the cytosolic TRC pathway component SGTA is cotranslationally recruited to ribosomes, which synthesize a diverse range of MPs, including those with cleavable SP [99].

The overlap in Snd2, plus the Wrb client spectra, determined after simultaneous depletion, is as follows: 30.2% precursors with SP (including 11.6% precursors of type I MPs and 4.7 precursors of multispanning MPs) and 69.8% precursors of MPs with TMH (Appendix A). Thus, 86% of the Snd2 and Wrb clients were precursors of MPs. The precursors with TMH included 48.8% multispanning MPs, 7% type II and type III MPs, 11.6% TA proteins, and 2.3% HP proteins (Appendix A). When the TMHs were analyzed for their positioning within the precursor proteins, 60% turned out to have comparatively amino-terminal TMHs (Appendix A). Furthermore, the client spectrum overlap demonstrated a lower preference for N-terminal topogenic sequences (72.1%) as compared to SR (87%) and a similar one as compared to SND (71%) and TRC/GET (70%). Thus, this analysis confirmed the strong overlap between the client spectra of the SND and TRC/GET pathways, which is consistent with the observations on these two pathways in yeast cells [74].

#### 2.2.4. The PEX3-Dependent Targeting Pathway

Recently, the PEX19/PEX3 pathway was characterized as yet another pathway for the targeting of precursor polypeptides to the ER and their subsequent insertion into the ER membrane [72,73]. PEX3 had originally been characterized as a peroxisomal membrane protein, which co-operates with the cytosolic protein PEX19 and the peroxisomal MP PEX16 in the targeting of peroxisomal MPs to pre-existent peroxisomes and in facilitating their membrane insertion [7,8]. So far, however, it remains to be elucidated whether or not these three PEX proteins are sufficient to insert any kind of MPs into the peroxisomal membrane [7]. As it turned out, PEX3 is also present in ER subdomains, which may be identical to the pre-peroxisomal ER and involved in the targeting of certain precursor proteins to ER membranes and in their membrane insertion [72,73]. These precursor proteins include HP proteins, which either remain in the ER or are pinched off with lipid droplets [72,73]. Notably, other HP proteins were observed to involve the GET pathway in yeast (Erg1) [141] and EMC in the human system [100]. Together, these observations raised the question of whether this pathway also plays a more general role in protein targeting to the ER [7,8]. Therefore, we addressed the client spectrum of PEX3 in ER protein targeting in human cells and, thereby, asked if the PEX19/PEX3 pathway to the ER can indeed target precursors to the Sec61 complex. Here, the approach involved chronically PEX3-deficient Zellweger patient fibroblasts [139].

Indeed, the PEX3 client spectrum are as follows: 56% precursors with SP (including 10% precursors of type I MPs, plus 2% precursors of multispanning MPs) and 44% precursors of MPs with TMH, including five peroxisomal MPs with TMH (Figure 4D, Appendix A). Thus, 56% of the PEX3 clients were precursors of MPs. The precursors with TMH included 18% type II and type III MPs, 14% multispanning MPs 8% TA proteins, and 4% HP proteins (Figure 4D, Appendix A). Thus, the PEX3 client spectrum did not show a preference for multispanning MPs (16%), as compared to 31.5% for SR, 50.9 for SND, and 44.3% for TRC/GET. When the TMHs were analyzed for their positioning within the precursor proteins, 46% turned out to have central or even C-terminal TMHs (Figure 5D, Appendix A), as compared to 24% for SR, 42% for SND, and 47% for TRC/GET. The presence of HP and peroxisomal MPs confirmed the feasibility of the experimental approach. However, the PEX3 client spectrum also demonstrated an unexpected preference for amino-terminal topogenic sequences (80%), which is similar to SR (87%), including 27 proteins with cleavable SP, such as five collagens. In addition, a strong overall bias of PEX3 (16%) against multispanning MPs became apparent (Figure 4D, Appendix A). To possibly explain the first of these latter two phenomena, we proposed that the PEX3 subdomain with its enrichment of HP proteins, including TANGO, creates an environment which also attracts collagens and may be conducive to the budding of peroxisomal precursors and large cargo secretory vesicles, as well as the formation of LDs [139].

### 2.3. Insertion of Precursor Polypeptides into and Translocation across the Membrane of the Human Endoplasmic Reticulum

#### 2.3.1. The Sec61 Complex as Central Entry Point into the ER

The heterotrimeric Sec61 complex provides the general entry point for precursor polypeptides with SPs or TMHs into the ER. Following their co- or posttranslational membrane targeting, the SPs or TMHs of precursor polypeptides are handed over to the Sec61 complex and begin to sample the cytosolic funnel of the Sec61 channel and to trigger the opening of the channel, which subsequently allows both the translocation of large hydrophilic domains of MPs (i.e., with more than 50 amino acid residues) or entire soluble proteins via the aqueous channel pore into the ER lumen, or the membrane insertion of TMHs or additional TMDs into the phospholipid bilayer via the lateral gate of the fully open Sec61 channel (Figure 1 and Figure 2) [142,143,144]. As mentioned above, the depletion of Sec61α served as a proof of principle of the approach for the components, which are involved in the insertion of precursor polypeptides into the ER membrane and their translocation into the ER.

The Sec61α depletion caused the degradation of the other two subunits of the heterotrimeric Sec61 complex, Sec61β and Sec61γ [135]. Thus, the cells were depleted of the entire Sec61 complex. The Sec61 client spectrum are as follows: 68.9% precursors with SP (including 23.8% precursors of type I MPs and 3.1% precursors of multispanning MPs) and 31.1% precursors of MPs with TMH (Figure 6A, Appendix A). Thus, 58% of the Sec61 clients were precursors of MPs. The precursors with TMH included 14% multispanning MPs, 15.4% single-spanning type II and type III MPs, and 1.7% TA proteins (Figure 6A, Appendix A). There were no HP proteins found among the Sec61 clients. When the TMHs were analyzed for their positioning within the precursor proteins, 77% turned out to have comparatively amino-terminal TMHs (Figure 7A, Appendix A). Thus, the Sec61 client spectrum demonstrated a clear overall preference for amino-terminal topogenic sequences (92%) and a slight overall bias against multispanning MPs (17.1%) (Figure 6A, Appendix A).

#### 2.3.2. The Auxiliary Sec62/Sec63 Complex with or without Help from BiP

The Sec61-associated Sec62/Sec63 heterodimer supports co- and posttranslational ER protein import in a client-specific manner and with or without support from the ER lumenal chaperone BiP [53,54,55,56,57,58,136,145,146]. We suggested that low hydrophobicity extends the dwell time of SPs at the cytosolic funnel of the Sec61 channel and that, therefore, auxiliary components of the Sec61 channel have to lower the activation energy for channel opening by direct interaction, particularly when aberrant SP hydrophobicity is combined with low SP helix propensity, as in the case of TRAP-dependent precursors (see below), or with deleterious features downstream of the SP, as in the Sec62/Sec63-dependent [135,136] case. Interestingly, both auxiliary complexes bind to ER lumenal loop 5 of Sec61α [46,59,60,61,62]. This interaction may support the rigid body movement in the course of Sec61 channel gating to the open conformation (Figure 1). In tthe case of additional BiP involvement in channel opening, it is recruited to the Sec61 complex by Sec63 and binds to ER lumenal loop 7 of Sec61α, thus contributing to the channel opening [75,146]. Furthermore, in both cases, a chaperone component of the auxiliary complexes (i.e., BiP in the complex, together with Sec62/Sec63 and the α-subunit in the TRAP complex) can bind to the soluble polypeptides or domains in transit and support translocation by acting as molecular ratchets and/or support folding of these polypeptides or domains [46,47,52,147].

The Sec62 client spectrum is as follows: 73.3% precursors with SP (including 16.8% precursors of type I MPs and 3.9% precursors of multispanning MPs) and 26.7% precursors of MPs with TMH (Figure 6C, Appendix A). Thus, 47.5% of the Sec62 clients were precursors of MPs. The precursors with TMH included 14.9% multispanning MPs and 11.9% type II and type III MPs (Figure 6C, Appendix A). There were no HP and TA proteins found among the Sec62 clients. When the TMHs were analyzed for their positioning within the precursor proteins, 74% turned out to have comparatively amino-terminal TMHs (Figure 7C). Thus, the Sec62 client spectrum demonstrated a clear overall preference for amino-terminal topogenic sequences (93.1%), SPs in particular, and an apparent overall bias against multispanning MPs (18.8%) (Figure 6C, Appendix A).

The Sec63 client spectrum is as follows: 47.8% precursors with SP (including 16.4% precursors of type I MPs and 4.5% precursors of multispanning MPs) and 52.2% precursors of MPs with TMH (Figure 6D, Appendix A). Thus, 73.1% of the Sec63 clients were precursors of MPs. The precursors with TMH included 35.6% multispanning MPs, 12.1% type II and type III MPs, and 1.7% TA proteins (Figure 6D, Appendix A). There were no HP proteins found among the Sec63 clients. When the TMHs were analyzed for their positioning within the precursor proteins, 69% turned out to have comparatively amino-terminal TMHs (Figure 7D, Appendix A). Thus, the Sec63 client spectrum demonstrated a less pronounced preference for amino-terminal topogenic sequences (83.6%), as compared to Sec62 (93.1%), and a more pronounced overall preference for multispanning MPs (40.1) than Sec62 (18.8%) (Figure 6D, Appendix A).

The combined Sec62 and Sec63 client spectrum, i.e., the Sec62/Sec63 client overlap, is as follows: 63.3% precursors with SP (including 26.7% precursors of type I MPs but no precursors of multispanning MPs) and 36.7% precursors of MPs with TMH (Appendix A). Thus, 63.3% of the Sec62 and Sec63 clients were precursors of MPs. The precursors with TMH included 23.3% multispanning MPs and 13.3% type II and type III MPs (Appendix A). There were no HP and TA proteins found among the Sec62 plus Sec63 clients. When the TMHs were analyzed for their positioning within the precursor proteins, 73% turned out to have comparatively N-terminal TMHs (Appendix A). Thus, the Sec62 plus Sec63 client spectrum demonstrated an intermediate preference for amino-terminal topogenic sequences (90%) and an intermediate preference for preference for multispanning MPs (23.3%). The comparison of the three client spectra of Sec62, Sec63, and their overlap is consistent with the observation from in vitro studies that Sec62 and Sec63 are not always acting on precursor polypeptides in concert [57,58].

#### 2.3.3. The Auxiliary TRAP Complex

Originally, TRAP was characterized as a signal-sequence receptor (SSR) complex, was demonstrated to be associated with the Sec61 complex, and was cross-linked to nascent polypeptides at the late translocation stages in the human system [42,44,46,56,148]. There is no obvious TRAP ortholog in yeast. As mentioned in the introduction, the ribosome-associated Sec61 complex and the TRAP form two stable stoichiometric super-complexes, called the OSTA or TRAP translocon [44,46,81,148]. In addition, TRAP can be found in a transiently forming super-complex (the multipass-TRAP translocon), comprising the TRAP translocon plus the GEL-BOS complex [81]. In vitro transport studies showed that the TRAP stimulates protein translocation depending on the efficiency of the SP in transport initiation, and Sec61 gating efficiency and TRAP dependence were found to be inversely correlated [43]. More recent studies in intact cells suggest that TRAP may also affect TMH topology, which is reminiscent of Sec62/Sec63 in yeast [43,149].

The TRAPβ depletion caused the degradation of the other three subunits of the TRAP complex [135]. Thus, the cells were depleted of the heterotetrameric TRAP complex. The TRAP client spectrum is as follows: 54% precursors with SP (including 12.1% precursors of type I MPs and 5.6% precursors of multispanning MPs) and 46% precursors of MPs with TMH (Figure 6B, Appendix A). Thus, 63.7% of the TRAP clients were precursors of MPs. The precursors with TMH included 33.9% multispanning MPs, 10.5% single-spanning type II and type III MPs, and 1.6% TA proteins (Figure 6B, Appendix A). There were no HP proteins found among the TRAP clients. When the TMHs were analyzed for their positioning within the precursor proteins, 67% turned out to have comparatively amino-terminal TMHs (Figure 7B, Appendix A). Thus, the TRAP client spectrum demonstrated a preference for amino-terminal topogenic sequences (84.7%) and an overall preference for multispanning MPs (39.5%) (Figure 6B, Appendix A), which is in perfect agreement with the observation of the multipass-TRAP translocon [81]. Notably, the first is in full agreement with the observation of the multipass-TRAP translocon [81].

#### 2.3.4. The Auxiliary TRAM1 Protein

TRAM or TRAM1 represents a multispanning MP of the human ER with eight TMDs; there is no obvious yeast ortholog [63]. It belongs to a protein family with the so-called TLC homology domain, which may bind ceramide or related sphingolipids [137]. Like TRAP, it was originally discovered by the cross-linking of nascent presecretory proteins, but, in contrast to TRAP, early in their translocation into the ER [32,63]. Furthermore, it was found to interact with nascent MPs in the course of their initial integration into the Sec61 channel [64,65,66,67]. TRAM1 was also among the first ER protein import components found to provide client-specific support [62]. Furthermore, it was observed that precursor proteins with short charged amino-terminal domains in their SPs require TRAM1 [65,66] and that precursors with short H-regions require the support from TRAM1 for efficient insertion into the lateral gate (Figure 2) [137].

The TRAM1 client spectrum is as follows: 43.3% precursors with SP (including 6.7% precursors of type I MPs and 6.7% precursors of multispanning MPs) and 56.7% precursors of MPs with TMH (Figure 6E, Appendix A). Thus, 70% of the TRAM1 clients were precursors of MPs. The precursors with TMH included 16.7% multispanning MPs, 33.3% single-spanning type II and type III MPs, and 6.7% TA proteins (Figure 6E, Appendix A). There were no HP proteins found among the TRAM1 clients. When the TMHs were analyzed for their positioning within the precursor proteins, 71% turned out to have comparatively amino-terminal TMHs (Figure 7E, Appendix A). Thus, the TRAM1 client spectrum demonstrated a clear overall preference for amino-terminal topogenic sequences (83.3%), and the highest preference for single-spanning type II and type III MPs among the auxiliary components of the Sec61 channel for membrane insertion (33.3%) (Figure 6E, Appendix A).

#### 2.3.5. The ER Membrane Complex or EMC

As described in the introduction, the decameric EMC belongs to the Oxa1-related MP insertases and, depending on its clients, may act co- or posttranslationally. TA MPs are some of its clients in yeast, as well as in the human system; most of them, however, are multispanning MPs, which are cotranslationally inserted into the ER membrane in close co-operation with the Sec61 channel [126,138]. Notably, some single-spanning type III MPs, as well as some HP proteins, were also observed to involve the EMC in the human system [100,150].

According to our analysis, the EMC client spectrum is as follows: 18.6% precursors with SP (including no precursors of type I MPs and 3.4% precursors of multispanning MPs) and 81.3% precursors of MPs with TMH (Figure 6F, Appendix A). Thus, 84.7% of the EMC clients were precursors of MPs. The precursors with TMH included 76.3% multispanning MPs, 1.7% type II and type III MPs, and 3.4% TA proteins (Figure 6F, Appendix A). There were no HP proteins found among the EMC clients. When the TMHs were analyzed for their positioning within the precursor proteins, 73% turned out to have comparatively amino-terminal TMHs (Figure 7F, Appendix A). Thus, the EMC client spectrum demonstrated an overall preference for amino-terminal topogenic sequences (78%) and a pronounced overall preference (79.7%) for multispanning MPs (Figure 6F, Appendix A). In a previous analysis, we observed only little client overlap for the EMC and Sec61 complex [5]. There were eight precursors found in these experiments that showed a dependence on both complexes, five with SP and three with TMH. Among the precursors with SP, there was no MP, and the three precursors with TMH were musltispanning MPs (ANO6, ATP13A1, TMBIM6) with relatively amino-terminal TMHs and an N_in_ or type II membrane topology, according to the most advanced prediction tool for this purpose (https://dtu.biolib.com/DeepTMHMM/, accessed on 1 July 2023). Furthermore, we detected an overlap of six precursors of multispanning type II MPs between EMC and TRAP (ATP13A3, SLC4A2, SLC44A2, SOAT1, TMBIM6, TMEM199), which suggests that it is not only the Sec61 channel that co-operates with EMC in the biogenesis of certain MPs but the TRAP or OSTA translocon. This is consistent with the fact that several of the EMC clients are N-glycoproteins.

### 2.4. A Quantitaive Mass Spectrometry Approach for the Analysis of Protein Import into the Human ER Identifies Redundancies as well as Preferences of Import Components for Certain Clients

The additional analyses of the various clients with respect to the apparent ΔG values of their TMHs for membrane insertion (http://dgpred.cbr.su.se, accessed on 1 July 2023) and the numbers of ER lumenal domains with a content of >50 amino acid residues (https://dtu.biolib.com/DeepTMHMM/, accessed on 1 July 2023), respectively, led to the following results: With respect to the apparent ΔG values of their TMHs, the gradient in preferences went from more negative values (i.e., higher hydrophobicities) to more positive values (i.e., lower hydrophobicities) in the direction of PEX3 << Snd2 ≤ Wrb < SR for membrane targeting, and of Sec62 ≤ TRAM1 < Sec63 ≤ Sec61 ≤ TRAP < EMC, consistent with the known preference of EMC for TMHs with a lower hydrophobicity compared to other membrane protein insertases (Figure 8A) [70]. Taking *p* values into account, the ΔG values for Snd2 and Wrb clients were similar, as expected, but, surprisingly, more negative for PEX3 clients and more positive for SR clients, as compared to the other two targeting components (Figure 8A, Appendix A) [106]. Interestingly, for TA protein clients, the ΔG values were more negative for Snd2 clients, as compared to Wrb clients, and TRAM1 clients stood out in having the lowest hydrophobicity of all components (Figure 8B).

With respect to the numbers of large ER lumenal domains, PEX3 stood out among the targeting components by including 43% of MP precursors with TMHs, plus ER lumenal domains, with a content of >50 amino acid residues (Appendix A), which is in line with the observation that many PEX3 clients were found to comprise SPs, such as collagens, and to target these precursors to the Sec61 channel [139]. With respect to membrane insertion, EMC clients stood out with 42% (together with Sec62 clients with 58%), consistent with the fact that EMC co-operates with the Sec61 channel in the membrane insertion of clients with large ER lumenal domains [70]. Furthermore, the numbers were lowest for Snd2 (10%) and TRAM1 (0%), which would be consistent with the proposal that these two components may also have the ability to act as MP insertases for certain MP clients, such as TA proteins and single- or multispanning MPs with very small ER lumenal domains (see below).

## 3. Discussion

In the course of their biogenesis, approximately 30% of all human polypeptides have to enter the secretory pathway at the level of the ER. As outlined above, this process involves topogenic sequences at the level of the precursors and a whole variety of ER protein import components that guarantee ER targeting as well as membrane integration or translocation (Figure 9, Appendix A) [15,16,17]. The SRP/SR pathway mediates exclusively the cotranslational targeting of precursor polypeptides to the Sec61 channel in the ER membrane [2,18,19,20,21,22,23,24,25,26,27,28,29,30]. Alternatively, nascent as well as completely synthesized precursor polypeptides are targeted to the Sec61 complex by either the PEX3/19, SND, or TRC/GET pathway [3,57,58,71,72,73,74,75,87,88,89,90,91,92,93,94,95,96,97,98,99,100,101,102,103,104,105,106]. Subsequently, precursor polypeptides are typically inserted into or translocated across the ER membrane via the Sec61 channel [31,32,33,34,35,36,37,38,39,40,41,81,142,143,144]. For some precursors, however, auxiliary components of the Sec61 channel have to support the gating of the channel to the open state, such as Sec62/Sec63 [49,50,51,52,53,54,55,56,57,58,59,60,61,62,63,136], TRAM1 [64,65,66,67,137], or TRAP [42,43,44,45,46,47,48,135]. Additionally, it suffices to summarize from two recent reviews [69,70] that (i) the Sec61 channel typically facilitates the cotranslational membrane insertion of TMHs and downstream TMDs whenever hydrophilic flanking regions with more than 50 amino acid residues have to be translocated into the ER lumen and, for these activities, either acts on its own or in the form of supercomplexes, the TRAP translocon or the OSTA translocon (Figure 1, Appendix A) [81,117,118]; (ii) the Sec61 channel co-operates with the EMC by cotranslationally inserting TMDs of multispanning MPs that are located downstream of the TMH (Figure 2) [124,125,126,127,128,129,130,131]; (iii) the EMC can also insert into the ER membrane single-spanning type III MPs, as well as TA proteins (both containing hydrophilic domains with less than 50 amino acid residues that face the ER lumen); (iv) the EMC inserts into the ER membrane TMDs with comparatively low hydrophobicity; (v) the TRC/GET system targets to and inserts into the ER membrane TA proteins with relatively high hydrophobicity; (vi) the GET pathway can also target and insert HP proteins and GPI-anchored proteins [100,141]; (vii) the GEL-BOS-PAT complex can insert multispanning MPs into the ER membrane [121,122,123,127,131,132]; (viii) typically, the Sec61 complex is not directly involved in membrane insertion by the GEL-BOS-PAT complex but rather plays a passive role in both providing a ribosome-docking site, plus contributing to the formation of a lipid-filled cavity together with the back-of-Sec61 or BOS complex; (ix) the GEL-BOS-PAT complex with Sec61 was observed in native membranes by cryoelectron tomography as such, as well as in complex with TRAP, and termed multipass translocon and multipass-TRAP translocon, respectively, and may co-operate with the GEL-BOS-PAT complex in the biogenesis of multispanning MPs whenever large soluble domains (with a content of more than 50 amino acid residues) that flank TMDs have to be translocated into the ER [81,132]; and (x) the PEX3/19 and the SND pathways may also act as MP insertases for HP and/or TA proteins.

### 3.1. Client Spectra for and Characteristics of Components for Targeting of Precursor Polypeptides to and Insertion into the Membrane of the Human Endoplasmic Reticulum

Here, we compiled the results from several unbiased experiments that addressed the question of which precursor polypeptides are targeted to the ER membrane and inserted into the ER membrane in human cells under standardized conditions, where one component was depleted one-at-a-time, and the effects on the total cellular proteomes were analyzed by label-free quantitative MS and differential protein abundance analysis [106,126,135,136,137,139]. The complete datasets were described in the original articles and deposited to the ProteomeXchange Consortium with the dataset identifiers that are given in Appendix A [106,126,135,136,137,139].

As previously described, the global unbiased analysis confirmed the overlaps and redundancies in the targeting of precursor polypeptides to the human ER; i.e., all four known targeting pathways were found to be able to target topogenic sequences to the Sec61 complex (Figure 9) [5,106]. As expected, there were no TA membrane proteins found among the SR clients and the SRP/SR-dependent pathway showed the preference for precursors with N-terminal SPs or relatively amino-terminal TMHs (Figure 9 and Figure 10). Specifically, the gradient in these preferences was SR > PEX3 > SND = TRC/GET (Figure 10A, Appendix A). In contrast to the SRP/SR- as well as the PEX3/PEX19-dependent pathways, TRC- and SND-dependent ER protein targeting showed the expected preference for multispanning membrane proteins, as well as for membrane proteins with central or carboxy-terminal TMHs (Figure 5, Figure 9 and Figure 10A) [106]. Here, the gradient in preferences for multipanning MPs was TRC/GET > SND > SR > PEX3 (Figure 10A, Appendix A). Previously, the double deletion in yeast of SND plus GET were found to be lethal and, thus, had first observed the functional overlaps in membrane targeting for TA proteins [74]. Furthermore, the suggested preference of PEX3 for HP proteins was confirmed, but extended to SRP/SR, and the expected preference of the SND and TRC/GET pathways for TA proteins was confirmed and also extended to PEX3 (Figure 4 and Figure 9, Appendix A). However, the results also included three completely unexpected findings, the presence of many precursors with SP, such as collagens, among the PEX3 clients [139] (discussed in reference 5), and the presence of multispanning MPs among the TRAP and the Wrb/Caml clients [106], respectively, the latter of which pointed towards the possibility of a more general targeting role of the TRC/GET pathway than previously anticipated, but had already been suggested on the basis of classical import studies [99] (see in section 3.3 in the context of related human diseases). The additional analyses of the various clients with respect to the apparent ΔG values of their TMHs for membrane insertion (http://dgpred.cbr.su.se, accessed on 1 July 2023) also confirmed previous results, most notably that the hydrophobicity of the TMHs of TA proteins (the TA) is typically higher, i.e., that the apparent ΔG for membrane insertion is lower or more negative as compared to TMHs in general (Figure 8, A versus B), and that the apparent ΔG values of TA protein clients of EMC are less hydrophobic, i.e., have a more positive apparent ΔG as compared to the Wrb clients (Figure 8B) [70].

With respect to the components for the insertion of the precursors of MPs into the human ER membrane, the results on the auxiliary components of the Sec61 channel from the classical studies were confirmed; i.e., all three were found to share most of their client profiles with the Sec61 complex (Figure 9, compare client spectra for Sec61 channel, TRAM1, TRAP, and Sec62/Sec63) [135,136,137]. The gradient in preferences for N-terminal SPs or more amino-terminal TMHs was Sec62 > Sec61 > TRAP > Sec63 = TRAM1 > EMC, and the gradient in preferences for multipanning MPs was EMC >> Sec63 = TRAP > Sec62 > Sec61 > TRAM1 (Figure 10A, Appendix A). In addition, TRAM1 showed a striking preference for single-spanning type II and type III MPs, while EMC showed the opposite, i.e., a striking bias against these two types of MPs (Figure 6E,F and Figure 10B). In addition, TRAM1 also showed an unexpected preference for TA proteins; i.e., the gradient in preferences of MP insertases for TA proteins was Wrb >> TRAM1 >> Sec61 > EMC > TRAP > Sec63, and none of these components showed a preference for HP proteins (Figure 9, Appendix A). Previously, the double deletion in yeast of EMC and GET were found to be lethal and, thus, had first observed the functional overlaps in membrane insertion for TA proteins [70]. These findings raise the two intriguing questions of which MP insertase(s) handle(s) HP proteins with respect to membrane insertion and what makes TRAM1 particularly relevant to the membrane insertion of single-spanning type II and type III MPs, as well as TA proteins. The most likely answer to the first question appears to be that PEX3, possibly in combination with PEX16, may the major MP insertase for HP proteins. A possible answer to the second question may be that TRAM1’s TLC homology domain, which is supposed to bind ceramide and related sphingolipids, and its suggested position opposite of the lateral gate of the open Sec61 channel may be particularly conducive for the membrane insertion of the TMHs of these particular clients, possibly forming a lipid-filled cavity in analogy to the GEL-BOS-PAT complex. On an even more speculative note, TRAM1 may be able to act as an MP insertase for TA and single-spanning type II and type III MPs, in addition to its auxiliary role for the Sec61 complex. Notably, there are no data from global approaches on the clients of the GEL-BOS-PAT complex so far, but some of these may be among the clients of the Sec61 complex and TRAP, respectively, which were discussed here.

### 3.2. Limitations of the Experimental Approach

The latter notion already points out a first limitation of the MS approach and highlights the fact that it should, ideally, be complemented by proximity-labeling experiments, such as described for EMC or the alternative Sec61 complex in yeast [68,126]. Additional weaknesses of the approach are that, for technical reasons, there were no small presecretory proteins and small MPs among the various clients and that, for unknown reasons, there were no GPI-anchored MPs detected. In addition, we attribute both to the relatively low number of small and GPI-anchored proteins [53,75]. Overall, the latter two technical issues may have been responsible for the fact that the client spectra of Snd2 and Sec62 did not show the expected overlap (Appendix A). Furthermore, the experimental design prior to the MS analysis had its limitations. While the depletion time of 96 h was apparently appropriate for all the above-described experiments, expected, as well as unexpected, problems were encountered for two experiments, which did not make it into the present compilation. In order to address the role of the ER lumenal molecular chaperone BiP in supporting the Sec62/Sec63 complex in ER protein import, which had been deduced from classical experiments [49,52,57,136], BiP depletion was also studied by the MS approach [136]. Since BiP is also involved in protein folding and assembly in the ER (Figure 1) [5,6,146], the depletion time was shortened to 72 h in HeLa cells [136]. Even under conditions of only 75% BiP depletion, there were dramatic effects on the proteome, which were visible as the negligible enrichment of GO terms for endocytosis and exocytosis for the negatively affected proteome, and there was no enrichment of precursor proteins with SP, N-glycosylation, or membrane location. Additionally, the problems were indicated by the activation of the unfolded protein response [136]. Furthermore, a similar problem was encountered for a different combination of siRNAs targeting Snd2 and Wrb compared to the experiment, which is represented by Appendix A. In both cases, the GO analysis was enough to exclude the experiments from any further analysis. In addition, there was an MS experiment which addressed the client spectrum of ERj1, which had previously been proposed to be a functional homolog of the Sec62/Sec63 complex in human cells (as well as vertebrates in general) [155,156,157,158]. Here, the problem was that the negative effect on the relevant proteome was too small to warrant any firm conclusions about the role of ERj1 in ER protein import (n = 15, including 7 with SP and 8 with TMH) [112].

### 3.3. Open Questions

The above summarized redundancies in pathways for the targeting of precursors of membrane proteins to and subsequent insertion into the ER membrane raise the questions of why they evolved and what the possible benefits are. To begin with the first question, it is clear that the variations in features of MPs that need to be targeted to and inserted into the ER membrane are enormous and demand highly specific machineries to avoid mistargeting of both the ER-destined MPs to other organelles and the MPs, which need to enter other organelles, to the ER (Figure 2). Therefore, the machineries may have to be equally complex and, apparently, became more and more complex in the course of evolution, as the clients became more and more complex (Figure 3) [69,70]. This may explain why there are no obvious homologs of TRAM1, TRAP, or the GEL-BOS-PAT complex in yeast and why the human Sec62 (as well as vertebrate Sec62 in general), compared to its yeast counterpart, gained a ribosome-binding site during evolution and, thus, the ability to support not only posttranslational but also cotranslational ER protein import [54,55,56,57,58,159]. Addressing a second aspect, we suggested previously [136] that specific client features may allow the differential regulation of ER protein import under different cellular conditions by the phosphorylation and/or Ca^2+^ binding of ER import components, such as Sec62/Sec63 and TRAPα (Figure 9) [151,152,153,160,161,162,163].

The detected variations in the characteristics of topogenic sequences may also be responsible for the known precursor-specific defects and, therefore, organ specificities in various inherited human diseases (Appendix A), recently reviewed by Sicking et al. [6]. The diseases include *SEC61A1*-linked common variable immunodeficiency, neutropenia and tubulointerstitial kidney disease [164,165,166,167,168], *SEC61B*- and *SEC63*-linked polycystic liver disease [54,169,170,171,172,173,174,175,176,177], and *TRAP*-, as well as *TRC35*-, *TRC40*-, and *CAML*-linked congenital disorders of glycosylation [46,135,178]. Notably, the observation that Sec63 has a preference for multispanning MPs (seen in Figure 6D) is consistent with the fact that the two previously identified clients of Sec63 that are responsible for the disease phenotype of disturbed planar cell polarity in *SEC63*-linked polycystic liver disease, polycystins 1 and 2, are multispanning MPs in the plasma membrane [168,169,177]. Interestingly, the present analysis identified yet another multispanning plasma membrane protein, VANGL2 [179,180,181,182,183,184,185,186,187], that is involved in the establishment and regulation of planar cell polarity as a Sec63 client and, when absent, may also contribute to the phenotype of *SEC63*-linked polycystic liver disease (Appendix A). Similarly, the observation that Wrb has a preference for not only TA but also other proteins, including N-glycosylated proteins (seen in Figure 4C and Appendix A), is consistent with the fact that the defects in the TRC pathway are linked to congenital disorders of glycosylation [178], as had previously been observed for defects in TRAP [46,188,189].

In addition, there remain many questions on the targeting and insertion/translocation components, which suggest many additional experiments. This includes the potential roles of PEX19/PEX3, Snd2/Snd3, and TRAM1 as client-specific MP insertases, which need to be addressed with the purified and reconstituted components in the classical in vitro assays. These same assays will have to solve the puzzle of how multispanning MPs such as ITPR 1 and 3, with a domain organization of a large cytosolic amino-terminal domain (with a content of >two thousand amino acid residues), five downstream TMDs, a large ER lumenal domain (with a content of >100 amino acid residues and an N-glycosylation site), a relatively carboxy-terminal TMD, plus a cytosolic domain with >100 amino acid residues are targeted to and inserted into the ER membrane by SRP or Wrb and Sec61 and/or EMC, respectively. Furthermore, the potential involvement of PEX16, the nature of the elusive Snd1 homolog, and the 3D structure of the Snd2/Snd3 complex need to be solved. Other open questions are related to the potential homologs of Snd3 and TRAM1, i.e., BRI3BP and TRAM2 [106,190,191]. Yet another white area on the map of components involved in protein targeting to the ER, as depicted in Figure 3, concerns the role of the receptors for mRNAs (such as KTN1) and RNCs (such as RRBP1, LRRC59, and AEG-1) in the ER membrane [29,111,112,113,192,193,194,195,196,197,198,199,200,201,202,203,204,205,206,207,208,209,210,211,212,213,214,215,216,217,218]. Last but not least, there is hardly any information on the possible variations in the tissue distributions of the various targeting and membrane insertion components.

Speaking more generally on mass spectrometry methods, we wonder if some of the interactors of protein transport components that were identified by high-throughput approaches such as affinity capture MS [219,220,221], proximity labeling in combination with MS [222,223,224,225,226,227,228,229], or cross-linking MS [230,231,232] and reported in databases such as https://thebiogrid.org or https://www.proteinatlas.org, accessed on 1 July 2023, are, in fact, transiently interacting transport substrates of the proteins of interest. We are raising this question because we found some of the clients, which were described here, in these databases, but we are well aware of the fact that the argument also works in the other direction.

## 4. Materials and Methods

The experimental approach was developed to identify substrates or clients of components which are involved in targeting of precursor polypeptides to and/or their translocation/membrane insertion into the human ER under cellular conditions [135]. The unbiased approach represents a combination of (i) siRNA-mediated knock-down or CRISPR/Cas9-mediated knock-out of a certain component in human cells (HeLa or HEK293 cells); (ii) label-free quantitative mass spectrometric (MS) analysis of the total cellular proteome; and (iii) differential protein abundance analysis for two different cell pools that had been treated with two different siRNAs, which target the same mRNA, compared to a pool of cells, which had been treated with a non-targeting or control siRNA. In the case of knock-out cells, only two cell pools were compared, a control cell line and the knock-out line; alternatively, deficient patient fibroblasts were analyzed (as in the case of congenital disorders of glycosylation or Zellweger syndrome) [106,112,126,135,136,137,138,139]. Typically, efficient knock-down or knock-out were confirmed by Western blot, as well as label-free quantitative proteomic analysis, and shown to have only marginal effects on cell growth and viability. These results are shown in the respective original manuscripts [106,112,126,135,136,137,139]. The details of the label-free quantitative proteomic analysis were also described previously and are only briefly described below. Routinely, the original MS data were deposited to the ProteomeXchange Consortium (http://www.proteomexchange.org, accessed on 1 April 2023) with the dataset identifiers that are given in Appendix A.

### 4.1. Label-Free Quantitative Proteomic Analysis

As described previously, ‘1 × 10^6^ cells (corresponding to roughly 0.2 mg protein) were harvested after growth for 96 h, washed twice in PBS, and lysed in a buffer containing 6 M GnHCl, 20 mM tris(2-carboxyethyl)phosphine (TCEP; Pierce^TM^, Thermo Fisher Scientific, Darmstadt, Germany), and 40 mM 2-chloroacetamide (CAA; Sigma-Aldrich, Taufkirchen, Germany) in 100 mM Tris, at pH 8.0 [135,136,137,139]. The lysate was heated to 95 °C for 2 min, and then sonicated in a Bioruptor sonicator (Diagenode, Seraing, Belgium) at the maximum power setting for 10 cycles of 30 s each. For a 10% aliquot of the sample, the entire process of heating and sonication was repeated once, and then the sample was diluted 10-fold with digestion buffer (25 mM Tris, pH 8, 10% acetonitrile). The protein extracts were digested for 4 h with Lysyl endoproteinase Lys-C (Wako Bioproducts, Fujifilm, Neuss, Germany, enzyme to protein ratio: 1:50), followed by the addition of trypsin (Promega, Heidelberg, Germany) for overnight digestion (at an enzyme-to-protein ratio of 1:100). The next day, a booster digestion was performed for 4 h using an additional dose of trypsin (enzyme-to-protein ratio: 1:100). After the digestion, a 10% aliquot of peptides (corresponding to about 2 µg of peptides) were purified via SDB-RPS StageTips [233], eluted as one fraction, and loaded for MS analysis. Purified samples were loaded onto a 50 cm column (inner diameter: 75 microns; packed with ReproSil-Pur C18-AQ 1.9-micron beads, Dr. Maisch HPLC GmbH, Ammerbuch, Germany) via the autosampler of the Thermo Easy-nLC 1000 (Thermo Fisher Scientific) at 60 °C. Using the nanoelectrospray interface, the eluting peptides were directly sprayed onto the benchtop Orbitrap mass spectrometer Q Exactive HF (Thermo Fisher Scientific) [234]. The peptides were loaded in buffer A (0.1% (*v*/*v*) formic acid) at 250 nL/min, and the percentage of buffer B was ramped to 30% over 180 min, followed by a ramp to 60% over 20 min, then 95% over the next 10 min, and maintained at 95% for another 5 min [136]. The mass spectrometer was operated in a data-dependent mode, with typical survey scans from 300 to 1700 m/z (resolution of 60,000 at m/z = 200). Up to 15 of the top precursors were selected and fragmented using higher-energy collisional dissociation (HCD) with a normalized collision energy value of 28 [136]. The MS2 spectra were typically recorded at a resolution of 17,500 (at m/z = 200). Typically, the AGC targets for the MS and MS2 scans were set to 3E6 and 1E5, respectively, within a maximum injection time of 100 and 25 ms for the MS and MS2 scans, respectively. Dynamic exclusion was enabled in order to minimize the repeated sequencing of the same precursor ions, and was set to 30 s [136]’.

### 4.2. MS Data Analysis

Typically, the raw data were processed using the MaxQuant computational platform [235]. The peak list was searched against Human Uniprot databases (typically, with an initial precursor and fragment tolerance of 4.5 ppm) and the proteins were quantified across the samples using the label-free quantification algorithm in MaxQuant as the label-free quantification (LFQ) intensities [236]. We note that LFQ intensities do not reflect true copy numbers because they depend not only on the amounts of the peptides but also on their ionization efficiencies; thus, they only served to compare the abundances of the same protein in different samples [234,235,236,237,238,239]. In this way, each siRNA experiment provided proteome-wide abundance data as LFQ intensities for three sample groups—one control (the control siRNA-treated) and two stimuli (down-regulation by two different targeting siRNAs directed against the same gene)—with each having three data points. The missing data points were generated by imputation, whereby we distinguished two cases [135]. In order to identify which proteins were affected by knock-down in siRNA-treated cells relative to the control siRNA-treated sample, we log2-transformed the ratio between siRNA and the control siRNA samples, and performed two separate unpaired *t*-tests for each siRNA against the control siRNA sample [135]. The *p* values obtained by the unpaired *t*-tests were corrected for multiple testing using a permutation-based false discovery rate (FDR) test. The proteins with an FDR-adjusted *p* value of below 5% were considered to be significantly affected by the knock-down of the targeted protein. The results from the two unpaired *t*-tests were then intersected for further analysis, meaning that the abundance of all of the reported candidates was statistically significantly affected in both siRNA-silencing experiments. For completely missing proteins lacking any valid data points, the imputed data points were randomly generated in the bottom tail of the whole proteomics distribution, following the strategy in the Perseus software version 1.5.2.6 (http://maxquant.net/perseus/, accessed on 1 January 2019) [238]. For proteins with at least one valid MS data point, the missing data points were generated from the valid data points based on the local least squares (LLS) imputation method [239]. The validity of this approach was demonstrated [135]. Subsequent to the data imputation, gene-based quantile normalization was applied to homogenize the abundance distributions of each protein with respect to the statistical properties. All statistical analyses were performed using the R package of SAM (https://statweb.stanford.edu/~tibs/SAM/, accessed on 1 January 2019) [240]. In the case of knock-out cells, only two cell pools were compared, a control cell line and the knock-out line.

### 4.3. Characterization of Precursor Polypeptides

The protein annotations of the SPs, TMHs, and N-glycosylation sites were extracted from UniProtKB entries using custom scripts [135]. The enrichment of the functional gene ontology annotations (cellular components and biological processes) among the secondarily affected proteins was computed using the GOrilla package [241]. Using custom scripts, we computed the hydrophobicity score and glycine/proline (GP) content of SP and TMH sequences [135]. A peptide’s hydrophobicity score was assigned as the average hydrophobicity of its amino acids according to the Kyte–Doolittle propensity scale (averaged over the sequence length) [242] or determined via an online tool (https://www.bioinformatics.org/sms2/protein_gravy.html, accessed on 1 July 2023). GP content was calculated as the total fraction of glycine and proline in the respective sequence [135]. ΔG_app_ values of SP and TMH were calculated with the ΔG_app_ predictor for TM helix insertion (http://dgpred.cbr.su.se, accessed on 1 July 2023). MP types were taken from UniProtKB or GeneCards (https://www.genecards.org, accessed on 1 July 2023), or were determined by employing the most advanced prediction tool for this purpose (https://dtu.biolib.com/DeepTMHMM/, accessed on 1 July 2023) [243,244,245]. Where indicated, the p values were determined by the Wilcoxon signed-rank test using the SPSS software (version 27; IBM Corporation, Armonk, NY, USA) (Appendix A).

## 5. Conclusions

The evaluation of quantitative MS data and the subsequent differential protein abundance analyses characterized the topogenic sequences of the membrane protein clients of four membrane-targeting and six membrane insertion components for the human ER. As expected, membrane protein (MP) precursors with cleavable amino-terminal signal peptides (SPs) or amino-terminal transmembrane helices (TMHs) were found to be predominant clients of SRP, as well the Sec61 complex, while precursors with more central or even carboxy-terminal TMHs were found to dominate the client spectra of the SND and TRC/GET pathways for membrane targeting in living human cells. Furthermore, the MP insertase EMC was confirmed to have a preference for multispanning MPs with weakly hydrophobic TMHs and to have (tail anchor) TA protein clients with less hydrophobic TAs as compared to the Wrb/Caml insertase. For the EMC clients with TMH, there was an overlap with Sec61 clients of three type II musltispanning MPs with large ER lumenal domains (in two of the cases) and with TRAP clients of six type II multispanning MPs (including one Sec61 client), which suggests that it is not only the Sec61 channel that co-operates with EMC in the biogenesis of certain MPs but the TRAP or OSTA translocon.

In addition, several novel insights were made that allow a couple of speculations. According to its client spectrum, TRAM1 may be able to act as an MP insertase for single- as well as multispanning type II MPs with weakly hydrophobic TMHs and with negligible ER lumenal domains, possibly by providing a lipid-filled cavity. Like the Wrb/Caml insertase, the Snd2/Snd3 complex may be able to facilitate membrane insertion for certain MPs, such as TA proteins, as well as multispanning MPs with negligible ER lumenal domains and comparatively negative ΔG for TMH membrane insertion (such as Sec62), possibly by involving a putative coiled-coil domain within the cytosolic carboxy-terminal domain, as well as a possible hydrophilic vestibule near the membrane surface of Snd2, which is both reminiscent of the Oxa1 superfamily insertases. It is tempting to visualize the particular clients as bitopic hairpin (HP)-like proteins, which is consistent with the observation that a monotopic HP protein was found among the clients of the SND pathway. Following these highly speculative insertase activities of Snd2/Snd3 and TRAM1, respectively, at least for TA protein clients, the ΔG values were more negative for Snd2 clients as compared to Wrb clients, and TRAM1 clients stood out in having the lowest hydrophobicity. Obviously, all these speculations will have to be addressed in future work with the purified and reconstituted components in the classical in vitro assays.

## Figures and Tables

**Figure 1 ijms-24-14166-f001:**
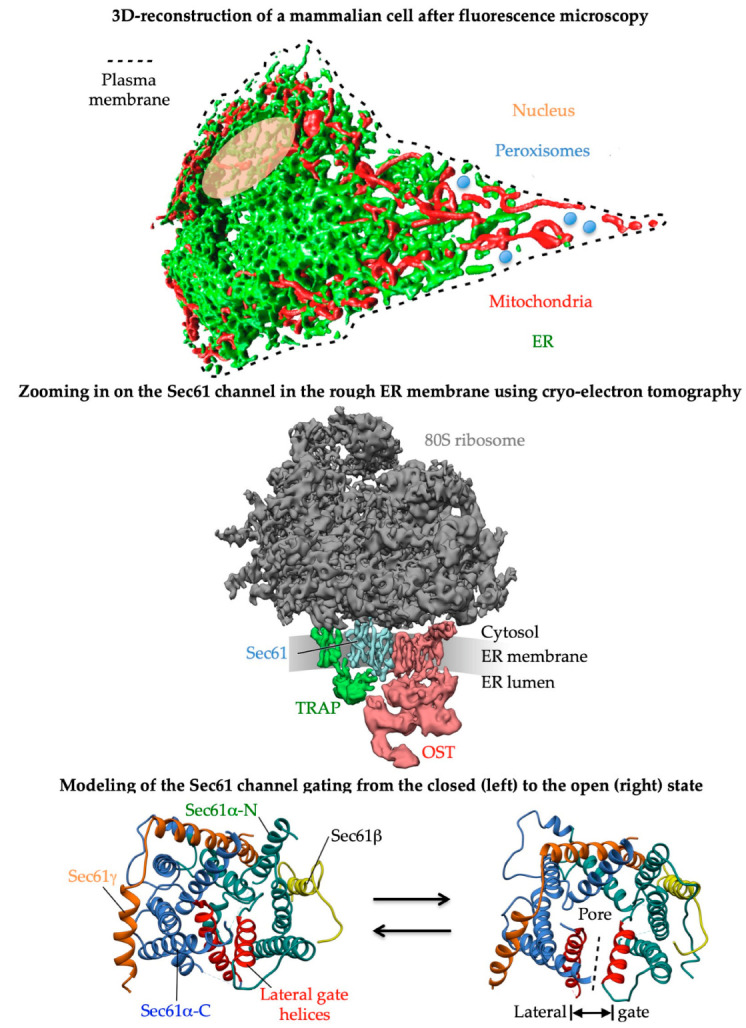
3D reconstructions of a typical nucleated human cell and a ribosome-bound Sec61 translocon. The figure and its legend were adapted from Lang et al. and Sicking et al. [5,6]. The upper part shows an artist’s view of a 3D reconstruction after live cell fluorescence imaging with ER-resident GFP and mitochondrial RFP and the central part a 3D reconstruction of the native ribosome-translocon complex in the human ER membrane after cryoelectron tomography. In human cells, the heterotrimeric Sec61 complex together with the ribosome form various large multicomponent complexes, e.g., the most abundant one comprising the multimeric membrane proteins translocon-associated protein (TRAP) and oligosaccharyltransferase (OSTA), which catalyzes N-linked glycosylation. This super-complex, now termed OSTA translocon, can insert into the membrane or translocate into the lumen a whole variety of topologically very different precursor polypeptides, i.e., type I-, type II-, glycosylphosphatidylinositol-, or GPI-anchored and type I multispanning membrane proteins, as well as soluble proteins, respectively (Figure 2). Membrane insertion and translocation are facilitated by either a cleavable amino-terminal SP or the TMH of the nascent precursor polypeptide, which acts as a non-cleavable SP substitute (Figure 2). The lowest part represents a modeling of reversible gating of the heterotrimeric Sec61 channel by SPs or TMHs. The fully open state of the Sec61 channel allows the translocation of hydrophilic domains of MPs or entire precursor polypeptides from the cytosol into the ER lumen (via the aqueous channel pore) and the insertion of transmembrane domains into the ER membrane (via the lateral gate), respectively.

**Figure 3 ijms-24-14166-f003:**
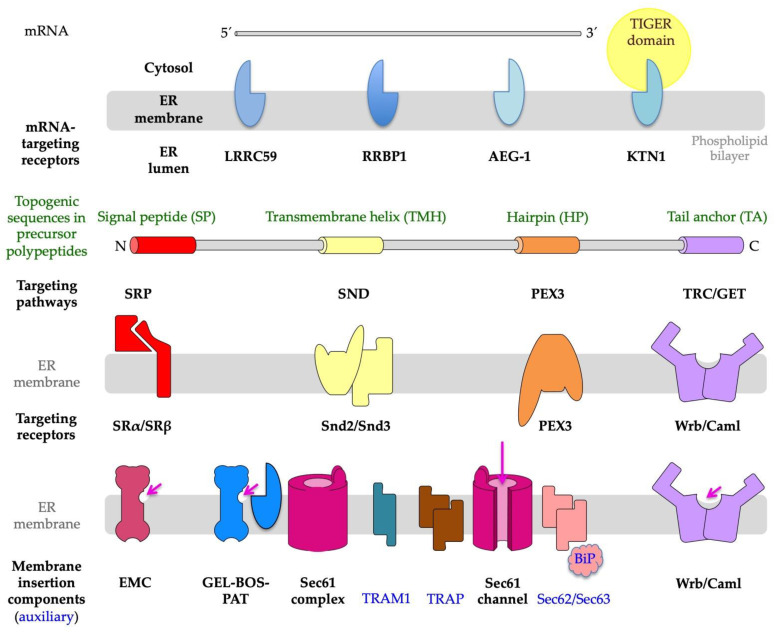
Components that are involved in protein import into the endoplasmic reticulum (ER) of human cells. ER protein import comprises a targeting step, plus a translocation or membrane insertion step, and may involve targeting of receptors for mRNAs or ribosomes with or without short nascent polypeptide chains to the ER (LRRC59, RRBP1, AEG-1, or KTN1). Alternatively, nascent or fully synthesized precursor polypeptides are targeted to the ER, depending on their topogenic sequences and targeting pathways, which involve cytosolic components, as well as their corresponding heterodimeric receptors in the ER membrane, such as SRα/SRβ, Snd2/Snd3, PEX3/PEX16, and Wrb/Caml. The membrane translocation is mediated by the heterotrimeric and polypeptide-conducting Sec61 channel, which may be supported by either the TRAP complex or the Sec62/Sec63 complex. Membrane insertion is mediated by several membrane protein insertases, i.e., (i) the Sec61 channel; (ii) the Sec61 channel with its partner complexes GEL, BOS, and PAT; (iii) the multimeric EMC; or (iv) the heterodimeric Wrb/Caml complex. The long arrow (in magenta) points to the open aqueous channel and open lateral gate, respectively, of the fully open Sec61 channel; the short arrows (in magenta) point to the characteristic hydrophilic vestibules of the MP insertases. Notably, (i) according to structural prediction tools, the yeast and human Snd2 may also be able to form a hydrophilic vestibule near the cytosolic surface of the ER membrane and, therefore, may also be able to facilitate membrane insertion [74,106]; (ii) the TIGER domain represents a cytosolic micro-domain, enriched in MP-encoding mRNAs with multiple AU-rich elements or AREs in their 3′UTRs in the vicinity of the ER [109,110]; and (iii) PEX3 is present in an ER subdomain which may be identical to the pre-peroxisomal ER [71,72,73]. The figure was adapted from Tirincsi et al. [106].

**Figure 4 ijms-24-14166-f004:**
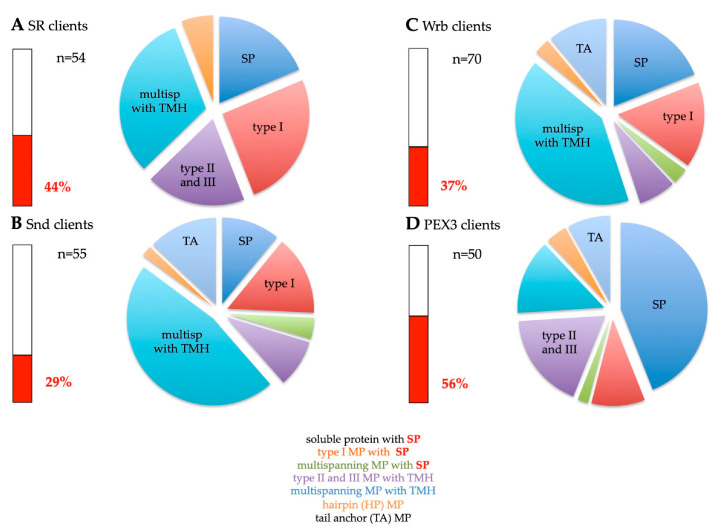
Distinguishing features of protein clients of the indicated components for targeting of precursor polypeptides to the ER. (**A**–**D**) The clients were determined by quantitative MS and differential protein abundance analysis following depletion of the respective component. Clients were defined as such by the presence of either an SP or at least one TMH. To characterize the clients of the various targeting and translocation or insertion components, the percentage of SP- (red bar) and TMH- (white bar) containing clients was calculated as given in Appendix A. The details of the client types were plotted in the pie diagrams as their relative distribution (colors are defined in the figure and are plotted clockwise starting from the twelve o’clock position in each pie). Original data are given in Appendix A. The figure was adapted in part from Tirincsi et al. [106].

**Figure 5 ijms-24-14166-f005:**
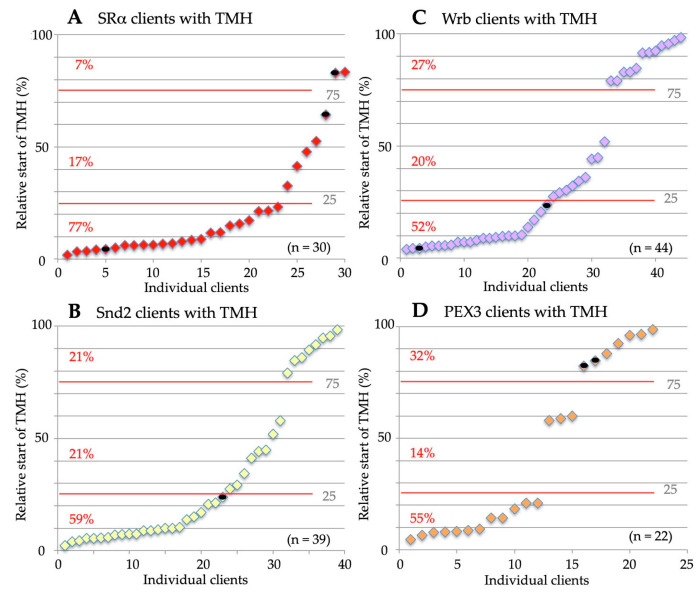
Distinguishing features of membrane protein clients of targeting components. (**A**–**D**) TMH-containing clients were plotted against the location of their TMH, i.e., position of central amino acid residue of TMH in % of client amino acid residues. For Wrb and hSnd2 clients, the data points and numbers (n) refer to the pooled clients (from the respective single depletion, hSnd2 or Wrb, and the double depletion, hSnd2 + Wrb) and are shown in Appendix A. Notably, the uppermost quarter includes MPs with TA, and the lowermost quarter those with rather N-terminal TMH; HP proteins are identified among clients by black dots. Furthermore, we note that Wrb is also part of a membrane insertase for tail anchor proteins, and that PEX3 may be a membrane protein insertase for HP proteins. The figure was adapted from Tirincsi et al. [106].

**Figure 6 ijms-24-14166-f006:**
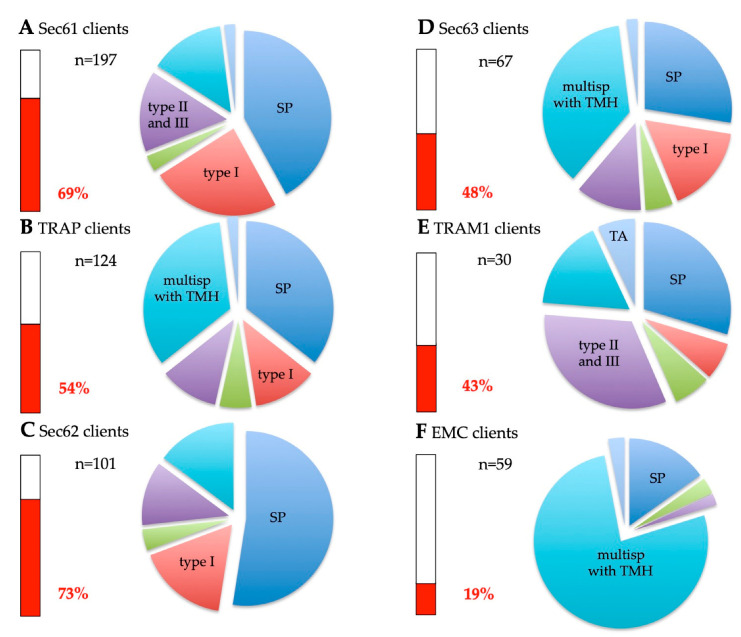
Distinguishing features of protein clients of the indicated components for translocation or membrane insertion. (**A**–**F**) The clients were determined by quantitative MS and differential protein abundance analysis following depletion of the respective component. Clients were defined as such by the presence of either an SP or at least one TMH. To characterize the clients of the various targeting and translocation or insertion components, the percentage of SP- (red bar) and TMH- (white bar) containing clients was calculated as given in Appendix A. The details of the client types were plotted in the pie diagrams as their relative distribution (colors are defined in the Figure and are plotted clockwise starting from the twelve o’clock position in each pie). Original data are given in Appendix A. The color code for the pies is shown in Figure 4.

**Figure 7 ijms-24-14166-f007:**
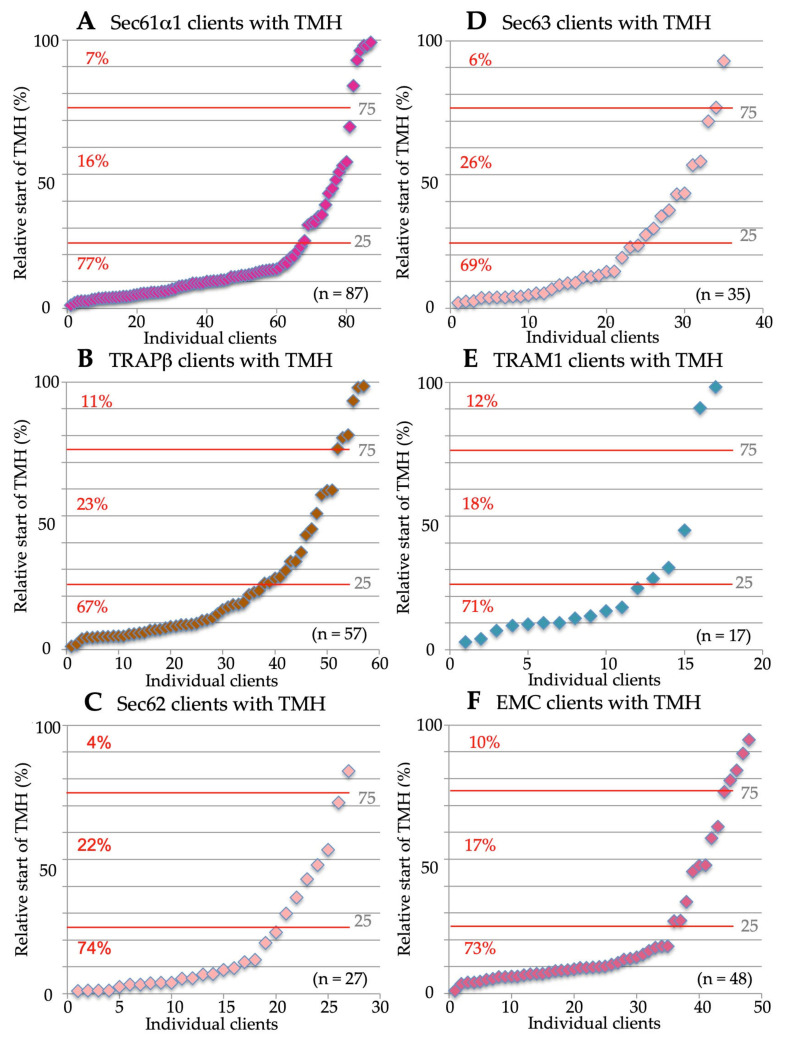
Distinguishing features of membrane protein clients of membrane insertion components. (**A**–**F**) TMH-containing clients were plotted against the location of their TMH, i.e., position of central amino acid residue of TMH in % of client amino acid residues. Notably, the uppermost quarter includes MPs with TA, and the lowermost quarter those with rather N-terminal TMH. Furthermore, we note that the data for the additional membrane protein insertase, comprising Wrb, are shown in Figure 6.

**Figure 8 ijms-24-14166-f008:**
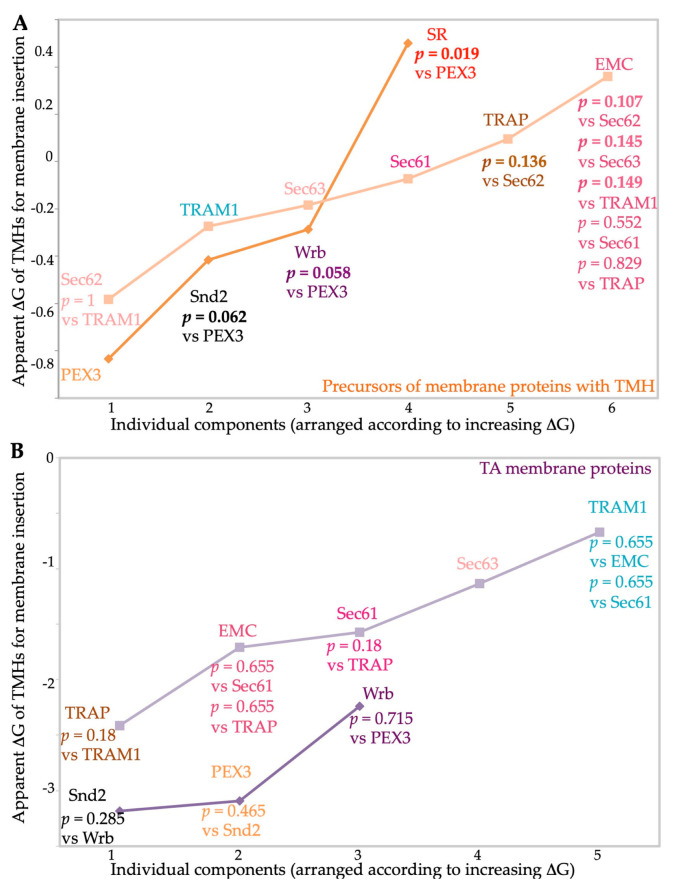
Apparent ΔG values for membrane insertion of TMHs from MP clients of components for targeting to and insertion into the human ER membrane. The average apparent ΔG values for membrane insertion for TMHs of clients of four targeting components (SR, Snd2, Wrb, and PEX3) and six membrane insertion components (Sec61, Sec62, Sec63, TRAMP, TRAM1, and EMC) were determined as described, taken from Appendix A, and comparatively shown together with selected *p* values, as determined by the Wilcoxon signed-rank test using the SPSS software (version 27; IBM Corporation, Armonk, NY, USA). (**A**) All precursors with TMHs are shown, including TA proteins. (**B**) Precursors of TA proteins are shown. Notably, there were no TA proteins among the clients of SR and Sec62, and only one in the case of Sec63; *p* values between 0.15 and 0.05 indicate tendencies towards significant differences and are highlighted in bold face; *p* values < 0.05 are considered to show significant differences and are also in bold face; vs, versus.

**Figure 9 ijms-24-14166-f009:**
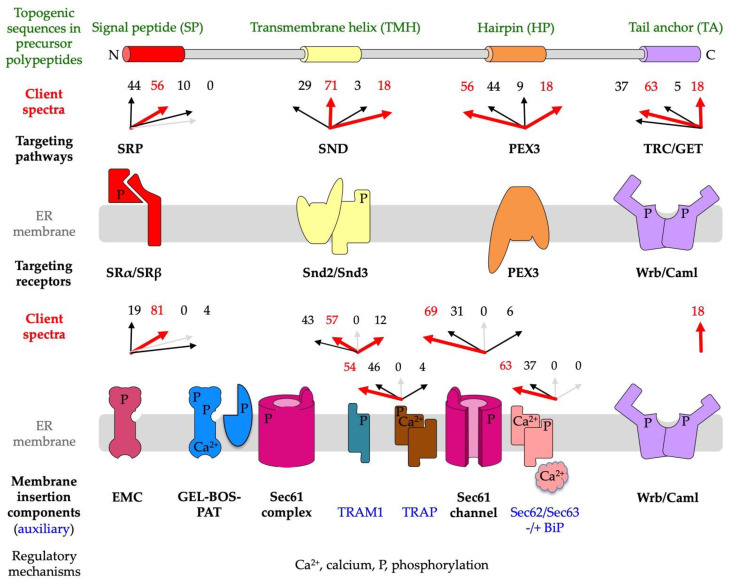
Components for protein targeting to and insertion into the human ER membrane with their client spectra and possible regulatory mechanisms. The client spectra were derived from the percentage data, which are depicted in Figure 4 and Figure 6; the first two percentages in each client spectrum refer to the percentage of precursors with SP and TMH {including HP and TA proteins), respectively, among the total number of clients, while the following two numbers refer to the percentage of precursors with HP or TA among the precursors with TMH. In the case of Sec62 and Sec63, the data refer to membrane protein clients that depend on Sec62, as well as Sec63 (Appendix A). The red arrows highlight top scoring numbers > 10, and light grey arrows represent the number 0, and are shown to indicate that fact. All numbers are given in Appendix A. The possible regulatory mechanisms were previously described in References [151,152,153,154], or found in databases.

**Figure 10 ijms-24-14166-f010:**
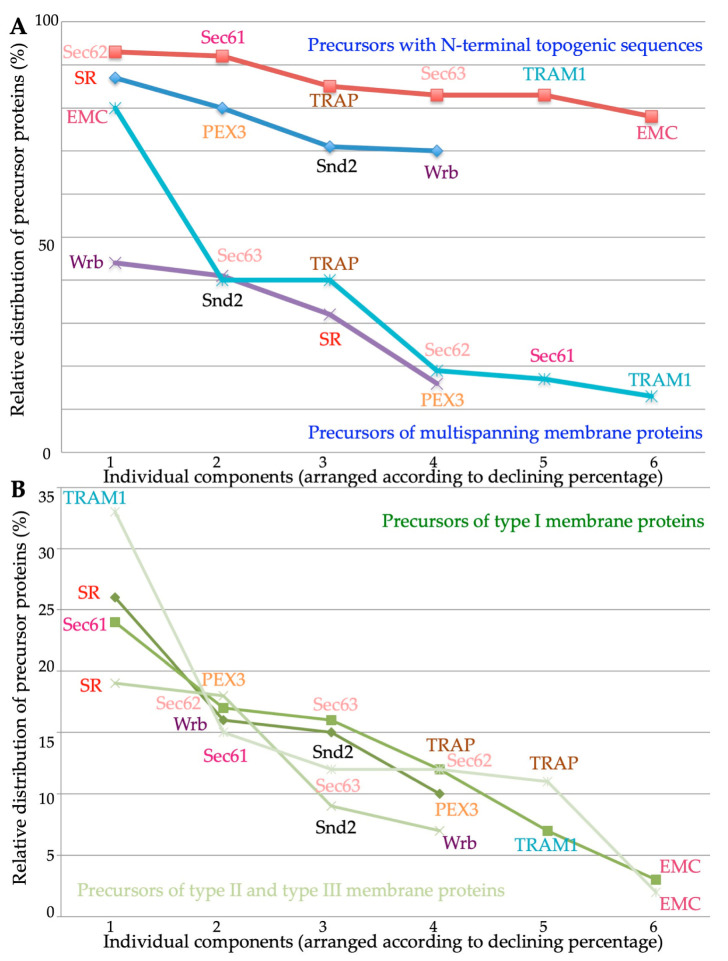
Client characteristics of components for targeting to and insertion into the human ER membrane. The client characteristics of four targeting components (SR, Snd2, Wrb, and PEX3) and six membrane insertion components (Sec61, Sec62, Sec63, TRAMP, TRAM1, and EMC) are summarized as relative distributions (i.e., relative to each other; given in % of the total number of precursor proteins of the respective component); i.e., the overall percentages of the particular types of precursor proteins were taken from Appendix A and Figure 4 or Figure 6 and comparatively shown. (**A**) Precursors with N-terminal topogenic sequences (SPs or TMHs) and precursors of multispanning MPs (irrespective of MP type, i.e., with SP, as well as with TMH) are shown. (**B**) Precursors of type I MPs, as well as precursors of single-spanning type II and type III MPs, are shown as indicated. Notably, type II and type III were not distinguished because the databases are currently rather incomplete in this respect, and HP and TA proteins are not represented here since they are shown in the client spectra of Figure 9.

## Data Availability

The original MS data (.raw and .txt files) have been deposited to the ProteomeXchange Consortium via the PRIDE partner repository with the indicated dataset identifiers (http://www.proteomexchange.org, accessed on 1 April 2023) as given in Appendix A. Any additional information required to analyze the data reported in this paper is available from the corresponding author upon request.

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
