# Peer review of "Quantitative Mass Spectrometry Characterizes Client Spectra of Components for Targeting of Membrane Proteins to and Their Insertion into the Membrane of the Human ER"

_ijms, 2023, doi:10.3390/ijms241814166_

Round 1

Reviewer 1 Report

This is an interesting paper addressing the specificity of ER protein targeting pathways that makes some important points about the latest mechanistic advances in the field.

The authors have taken a wide-ranging and meta analytical approach to examine knock-down of different components in HEK, HeLa and Zellweger patient cells. A caveat of this approach is that the cells are derived from different tissue types, and will be at different secretory capacity. The PEX3 deficient patient cells will also have significant peroxisome defects which will influence the metabolic poise of the cells and their overall proteostasis.

The paper is badged as a review but does contain some primary data analysis and meta analysis - as a result the initial pitch of the article is a little unclear.

In places, the experimental rationale was hard to follow and the materials and methods section was brief. Please could the authors include, for example, details of how the proteomics experiments were performed. Also, is experimental validation of the knock-downs and the label free quantitation available?

The proteomics data has been deposited in proteome exchange, but some information in the text is required to understand how the raw data has been processed, particularly for Fig 9 where relative distribution of precursors is shown – how does this relate to relative abundance, for example? How were the p values obtained for Fig 10?

The figures would benefit from some revision and reorganisation, and there is font inconsistency between the figures. Figure 2 could be better organised to fit with the legend on one page. The lettering of Fig 5 has rendered incorrectly – it would be easier to follow the figure if the figure were smaller and the legend were on the same page. For figures 1, 2, and 6 the legend text “on the following page” is not correct. In general the figure legends are a bit wordy- some text could be shortened or transferred to the main body of the paper.

On page 12, the text from ‘the approach relies … component by far’ appears to be quoted from a previous paper – if so, this should be reworded. The text in the discursive section is a bit fragmented and would benefit from some streamlining. The authors have tried to be comprehensive in citing the literature, but the paper is somewhat over referenced.

Minor points: in the introduction, it is a bit misleading to call the cytosol aqueous. There are some minor grammatical and spelling mistakes e.g. pg 11. Mediated.

The English is ok, but requires another edit to correct minor spelling and grammar issues and to make the paper more readable.

Author Response

Answers to Reviewer 1

This is an interesting paper addressing the specificity of ER protein targeting pathways that makes some important points about the latest mechanistic advances in the field.

We are gratefull for your assessment and your suggestions for improvement.

The authors have taken a wide-ranging and meta analytical approach to examine knock-down of different components in HEK, HeLa and Zellweger patient cells. A caveat of this approach is that the cells are derived from different tissue types, and will be at different secretory capacity. The PEX3 deficient patient cells will also have significant peroxisome defects which will influence the metabolic poise of the cells and their overall proteostasis.

We agree but cannot take this into account since cellular concentrations of proteins are known only for HeLa cells (among our models). We added to the methods section that the analyzed cells did not have any gross problems in cell growth or viability.

The paper is badged as a review but does contain some primary data analysis and meta analysis - as a result the initial pitch of the article is a little unclear.

This is a very valid point and was also brought up by the other reviewers. We have changed the manuscript to `Article´ but would be happy with whatever the Editor decides on this issue. Originally, we went for `Review´ since we do not report novel experimental data but only extended our analysis of the data, this time with the focus on membrane proteins.

In places, the experimental rationale was hard to follow and the materials and methods section was brief. Please could the authors include, for example, details of how the proteomics experiments were performed. Also, is experimental validation of the knock-downs and the label free quantitation available?

With the change of the manuscript to `Article´, we have extended the methods section and also clarified the issue of validation of knock-down efficiency.

The proteomics data has been deposited in proteome exchange, but some information in the text is required to understand how the raw data has been processed, particularly for Fig 9 where relative distribution of precursors is shown – how does this relate to relative abundance, for example? How were the p values obtained for Fig 10?

As stated above, cellular concentrations of proteins are known only for HeLa cells (among our models). We have pointed out in the legend to old Figure 9 (new 10) that `relative´ means relative to each other. Furthermore, we have added the required explanation to both the legend of old Figure 10 (new 8) and the respective methods section (4.3.). In addition, we have added the full data set as new Table S5.

The figures would benefit from some revision and reorganisation, and there is font inconsistency between the figures. Figure 2 could be better organised to fit with the legend on one page. The lettering of Fig 5 has rendered incorrectly – it would be easier to follow the figure if the figure were smaller and the legend were on the same page. For figures 1, 2, and 6 the legend text “on the following page” is not correct. In general the figure legends are a bit wordy- some text could be shortened or transferred to the main body of the paper.

We have improved the figures and eliminated the inconsistencies. Furthermore, we have redesigned the figures to make them fit on one page together with their legends.

On page 12, the text from ‘the approach relies … component by far’ appears to be quoted from a previous paper – if so, this should be reworded. The text in the discursive section is a bit fragmented and would benefit from some streamlining. The authors have tried to be comprehensive in citing the literature, but the paper is somewhat over referenced.

Our understanding was that pointing out the quotation can substitute for rewording but would be happy with whatever the Editor decides on this issue. We note that we have used the same strategy in the new methods section.

Minor points: in the introduction, it is a bit misleading to call the cytosol aqueous. There are some minor grammatical and spelling mistakes e.g. pg 11. Mediated.

We assume that you refer to the fact that the cytosol is gel-like and have added this to the text. We have tried to eliminate the grammatical and spelling mistakes but will also involve the MDPI English editors in further improvement (i.e. if the manuscript gets accepted).

Reviewer 2 Report

Although this is a comprehensive article about components targeting and inserting membrane proteins into the ER membrane and it could potentially be interesting, it is written in a very confusing way. It would have to be completely rewritten to be acceptable.

It is presented as a review, but then the title and abstract are of a research article. Then a long Introduction in the style of a review and then Results are shown and analyzed and presented in most cases in the style of a research article, making it very unclear for the reader whether this is original research or only analysis of previous work. 

The article should be presented as a comparative analysis of previous work, in the spirit of the beginning of section 3.1 “Here, we compiled the results from several unbiased experiments that addressed the question which precursor polypeptides are targeted to the ER membrane and insert ed into the ER membrane in human cells under standardized conditions where one component was depleted one at a time and the effects on the total cellular proteomes were analyzed by label free quantitative MS and differential protein abundance analysis…”

This should be the spirit of the title, abstract and along the text. The introduction should be drastically shortened into a normal introduction. This is not a Review, it is the further analysis of the data by the authors (and not a meta-analysis or review of the data from several research groups), with an addition of a Review styled Introduction. In addition, the Introduction/Review part is too similar to their published article Lang et al 2022.

Additionally, several of the figures are copies with or without minimal changes from Lang et al 2022 and Sicking et al 2021, such as Fig. 1, 2, 4 and Table I. Although this is cited in the figure legends, it is not sufficient for a new publication, which should have mostly new illustrations and insight.

The English is fine. Minor spelling and grammar check needed.

Author Response

Answers to Reviewer 2

Although this is a comprehensive article about components targeting and inserting membrane proteins into the ER membrane and it could potentially be interesting, it is written in a very confusing way. It would have to be completely rewritten to be acceptable.

We are gratefull for your assessment and your suggestions for improvement. We have reorganized the manuscript and rewritten it in parts.

It is presented as a review, but then the title and abstract are of a research article. Then a long Introduction in the style of a review and then Results are shown and analyzed and presented in most cases in the style of a research article, making it very unclear for the reader whether this is original research or only analysis of previous work. 

This is a very valid point and was also brought up by the other reviewers. We have changed the manuscript to `Article´ but would be happy with whatever the Editor decides on this issue. Originally, we went for `Review´ since we do not report novel experimental data but only extended our analysis of the data with the focus on membrane proteins.

The article should be presented as a comparative analysis of previous work, in the spirit of the beginning of section 3.1 „Here, we compiled the results from several unbiased experiments that addressed the question which precursor polypeptides are targeted to the ER membrane and insert ed into the ER membrane in human cells under standardized conditions where one component was depleted one at a time and the effects on the total cellular proteomes were analyzed by label free quantitative MS and differential protein abundance analysis…”

We had assumed that the definition of `meta-analysis´ would include this, but changed the wording as suggested.

This should be the spirit of the title, abstract and along the text. The introduction should be drastically shortened into a normal introduction. This is not a Review, it is the further analysis of the data by the authors (and not a meta-analysis or review of the data from several research groups), with an addition of a Review styled Introduction. In addition, the Introduction/Review part is too similar to their published article Lang et al 2022.

The Introduction was drastiacally shorthened and changed as suggested and the manuscript was changed to `Article´ (see above).

While most of the data were from our lab, we also included into our comparison similar work from two other labs on the membrane protein insertase, which is termed ER membrane complex (EMC), and stated as much in the Abstract.

Additionally, several of the figures are copies with or without minimal changes from Lang et al 2022 and Sicking et al 2021, such as Fig. 1, 2, 4 and Table I. Although this is cited in the figure legends, it is not sufficient for a new publication, which should have mostly new illustrations and insight.

We omitted old Figure 4 and moved Table 1 to the end of the Supplement as Table S6. This table is updated compared to previous versions and may be helpful to the reader.

Our understanding was that pointing out the quotation of cartoons can substitute for redrawing them but would be happy with whatever the Editor decides on this issue.

Reviewer 3 Report

I have carefully reviewed the manuscript titled "Quantitative mass spectrometry characterizes client spectra of components for targeting of membrane proteins to and their insertion into the membrane of the human ER" submitted for publication. Overall, the manuscript presents valuable insights into ER membrane proteins using mass spectrometry. Below, I have provided my revisions and comments for your consideration.

1. Figures 1, 4, 5, 6, 7, 9, and 10 should include properly labeled axes, units, and legends.

Redesign Figure 5 for improved clarity, adjusting word size, resolution, and placing explanatory text in the legend.

Ensure consistency in nomenclature, such as replacing "alphafold" in Figure 4.

Correct any missing information on the X axis of Figure 6 and 7.

2. Cite original publications for references [4]-[7] in Table 1 to enhance accessibility for readers.

3. Clarify the significance of ER membrane protein research and its implications in the field.

Provide a brief overview of the mass spectrometry method's importance in the study of ER membrane proteins.

4. Elaborate on the mass spectrometry technique used, including the specific protocols, instrument details, and data analysis procedures.

Introduce "classical analyses" in context and explain its relevance to the study.

5. Regarding Fig. 1A, I did not locate an explanation for the physiological role of calcium (Ca) in the manuscript. Additionally, could you please provide clarification on the reason for including Ca concentration labels on the image?

6.In 2.2.2, “consitent” should be "consistent'?

7. line 4 , should be "crispr"

8.In fig.2 I am confuse what A B and C indicated for. the picture looks like merge together.

9.When discussing the translocation of precursor membrane proteins through the Sec61 channel, it would be beneficial to include a reference to support the information. Despite the inclusion of the channel's open properties in Fig. 1, I believe that supplementary information is necessary for a more comprehensive understanding.

10. I feel its very difficult for reader to understand the Fig.9 and 10.

11. Not sure a review manuscript should include method and results section

Author Response

Answers to Reviewer 3

I have carefully reviewed the manuscript titled "Quantitative mass spectrometry characterizes client spectra of components for targeting of membrane proteins to and their insertion into the membrane of the human ER" submitted for publication. Overall, the manuscript presents valuable insights into ER membrane proteins using mass spectrometry. Below, I have provided my revisions and comments for your consideration.

We are gratefull for your assessment and your suggestions for revisions.

  1. Figures 1, 4, 5, 6, 7, 9, and 10 should include properly labeled axes, units, and legends.

Redesign Figure 5 for improved clarity, adjusting word size, resolution, and placing explanatory text in the legend.

Ensure consistency in nomenclature, such as replacing "alphafold" in Figure 4.

Correct any missing information on the X axis of Figure 6 and 7.

We apologize for the poor quality of some figures and have improved the figures and eliminated the inconsistencies. Furthermore, we have redesigned the figures to made them to fit on one page together with their legends. We have split old Figure 5 into new Figures 4 and 6 (omitting old Figure 4 upon request from another reviewer).

  1. Cite original publications for references [4]-[7] in Table 1 to enhance accessibility for readers.

We have moved Table 1 to the end of the Supplement as Table S6 upon request from another reviewer.

3a. Clarify the significance of ER membrane protein research and its implications in the field.

Seriously? Having asked that, we have added a statement towards this end to the Abstract (last sentence) as well as Introduction (represents a central cell biological research topic of the past fifty years as well as several years to come) and took it up in the Discussion by writing: `Interestingly, the present analysis identified yet another multispanning plasma membrane protein, VANGL2, that is involved in the establishment and regulation of planar cell polarity as Sec63 client and, when absent, may also contribute to the phenotype of SEC63-linked Polycystic Liver Disease (Table S4).´

3b. Provide a brief overview of the mass spectrometry method's importance in the study of ER membrane proteins.

We do not feel confident to do so and are convinced that with the change of the manuscript from `Review´ to `Article´ it would not be appropriate. However, we raised one question in the Discussion that may be of general interest to the MS community and is related to high-throughput methods for the detection of protein interactions. We wrote `Speaking more generally on mass spectrometry methods, we wonder if some of the interactors of protein transport components that were identified by high-throughput approaches such as proximity labeling and reported in databases, such as https://thebiogrid.org or https://www.proteinatlas.org are in fact transiently interacting transport substrates of the proteins of interest. We are raising this question because we found some of the clients, which were described here, in these databases but are well aware of the fact, that the argument also works in the other direction.´

4a. Elaborate on the mass spectrometry technique used, including the specific protocols, instrument details, and data analysis procedures.

With the change of the manuscript to `Article´, we have extended the methods section.

4b. Introduce "classical analyses" in context and explain its relevance to the study.

We introduced the term in the Introduction.

  1. Regarding Fig. 1A, I did not locate an explanation for the physiological role of calcium (Ca) in the manuscript. Additionally, could you please provide clarification on the reason for including Ca concentration labels on the image?

We omitted any reference to the role of BiP in maintaining the physiological calcium concentration in the ER lumen.

6.In 2.2.2, “consitent” should be "consistent'?

Thank you, it was corrected.

  1. line 4 , should be "crispr"

Thank you, it was corrected.

In general, we have tried to eliminate the grammatical and spelling mistakes but will also involve the MDPI English editors in further improvement (i.e. if the manuscript gets accepted).

8.In fig.2 I am confuse what A B and C indicated for. the picture looks like merge together.

Thank you, we took it out.

9.When discussing the translocation of precursor membrane proteins through the Sec61 channel, it would be beneficial to include a reference to support the information. Despite the inclusion of the channel's open properties in Fig. 1, I believe that supplementary information is necessary for a more comprehensive understanding.

We have added a statement to the Introduction that `TMDs can enter the phospholipid bilayer via the lateral gate of the fully open Sec61 channel by lateral movement and large hydrophilic domains (i.e. with more than 50 amino acid residues) or entire soluble proteins can be translocated into the ER lumen by vectorial movement through the channel (Figures 1 and 2).´

  1. I feel its very difficult for reader to understand the Fig.9 and 10.

We apologize for the poor quality of old Figures 9 and 10 (new 8 and 10) and have improved the figures. Furthermore, we have redesigned the figures to make them fit on one page together with their legends.

  1. Not sure a review manuscript should include method and results section

With the change of the manuscript to `Article´, we have extended the methods section (see aove).

Round 2

Reviewer 1 Report

Although the authors have addressed most of the points raised, I still have concerns that the use of brackets in the materials and methods is misleading, especially for a casual reader. It is not clear which methods have been primarily used for the meta data analysis in this paper, and which methods were used by authors of the original papers. Therefore I would like to see some rewriting of the materials and methods to make this aspect transparent.

Reviewer 2 Report

The manuscript is much improved. Although in my opinion the Introduction should be still shorter,  I believe that the article is now much clearer and now acceptable and a valuable addition to this field.

The English is fine. Minor spelling and grammar check needed.

Reviewer 3 Report

The author answered all my questions.